# Measuring spectrally-resolved information transfer

**Edoardo Pinzuti**[1]*, **Patricia Wollstadt**[2], **Aaron Gutknecht**[3], **Oliver Tüscher**[1,4], **Michael Wibral**[2,3]

**1** Leibniz Institute for Resilience Research, Mainz, Germany, **2** MEG Unit, Brain Imaging Center, Goethe University, Frankfurt/Main, Germany, **3** Campus Institute for Dynamics of Biological Networks, Georg August University, Göttingen, Germany, **4** Department of Psychiatry and Psychotherapy, Johannes Gutenberg University of Mainz, Mainz, Germany

* Edoardo.Pinzuti@lir-mainz.de

**Data Availability Statement:** All the code and a demo script are available at https://github.com/pwollstadt/IDTxl/tree/feature_spectral_te.

**Funding:** EP, OT, MW received support from the SFB 1193, C04, funded by the Deutsche

## Abstract

Information transfer, measured by transfer entropy, is a key component of distributed computation. It is therefore important to understand the pattern of information transfer in order to unravel the distributed computational algorithms of a system. Since in many natural systems distributed computation is thought to rely on rhythmic processes a frequency resolved measure of information transfer is highly desirable. Here, we present a novel algorithm, and its efficient implementation, to identify separately frequencies sending and receiving information in a network. Our approach relies on the invertible maximum overlap discrete wavelet transform (MODWT) for the creation of surrogate data in the computation of transfer entropy and entirely avoids filtering of the original signals. The approach thereby avoids well-known problems due to phase shifts or the ineffectiveness of filtering in the information theoretic setting. We also show that measuring frequency-resolved information transfer is a partial information decomposition problem that cannot be fully resolved to date and discuss the implications of this issue. Last, we evaluate the performance of our algorithm on simulated data and apply it to human magnetoencephalography (MEG) recordings and to local field potential recordings in the ferret. In human MEG we demonstrate top-down information flow in temporal cortex from very high frequencies (above 100Hz) to both similarly high frequencies and to frequencies around 20Hz, i.e. a complex spectral configuration of cortical information transmission that has not been described before. In the ferret we show that the prefrontal cortex sends information at low frequencies (4-8 Hz) to early visual cortex (V1), while V1 receives the information at high frequencies (> 125 Hz).

## Author summary

Systems in nature that perform computations typically consist of a large number of relatively simple but interacting parts. In human brains, for example, billions of neurons work together to enable our cognitive abilities. This well-orchestrated teamwork requires information to be exchanged very frequently. In many cases this exchange happens rhythmically and, therefore, it seems beneficial for our understanding of physical systems if we

Forschungsgeminschaft (DFG). EP is currently employed by the Leibniz Institute for Resilience Research funded by the Leibniz Gemeinschaft. The funders had no role in study design, data collection and analysis, decision to publish, or preparation of the manuscript.

**Competing interests:** The authors have declared that no competing interests exist.

could link the information exchange to specific rhythms. We here present a method to determine which rhythms send, and which rhythms receive information. Since many rhythms can interact at both sender and receiver side, we show that the above problem is tightly linked to partial information decomposition—an intriguing problem from information theory only solved recently, and only partly. We applied our novel method to information transfer in the human inferior temporal cortex, a brain region relevant for object perception, and unexpectedly found information transfer originating at very high frequencies at 100Hz and then forking to be received at both similarly high but also much lower frequencies around 20Hz. These results overturn the current standard assumption that low frequencies send information to high frequencies.

This is a *PLOS Computational Biology* Methods paper.

## Introduction

Many natural or artificial complex systems perform distributed computation. In a distributed computation multiple relatively simple parts of the system perform rather elementary operations on their inputs, but do communicate heavily amongst each other in order to jointly implement complex computations. Along similar lines other complex systems in a similar way consist of many interacting simple parts that exchange information in some sense. Thus, to understand such complex systems, measuring the information transferred between the parts of the system is crucial. A mathematically rigorous measure of information transfer is the transfer entropy (TE) [1]. TE, as a model-free information theoretic measure, is ignorant of the details on how the information transfer is physically implemented, which is indeed a highly desirable property when we only want to detect and measure information transfer. However, many systems display highly rhythmic activity when performing distributed computation, suggesting that measuring the information transfer associated with different spectral components may provide valuable additional insights. This holds in particular for biological neural systems where rhythmic or quasi-periodic activity is found frequently across many scales from spiking activity of individual neurons to electroencephalographic (EEG) recordings of large pools of neurons (see [2] and references therein).

Early attempts [3] to obtain the desired frequency-resolved measurement of TE resorted to narrow-band filtering of the data from information source and information receiving target and to feeding the resulting narrow-band signals into a TE analysis (including the additional calculation of a signal envelope in [3]). Yet, these approaches come with certain problems that are well-known from the field of Granger-Causality (GC) analysis. Due to the equivalence of GC and TE for jointly Gaussian variables [4], these problems carry over to TE analyses:

1. Most importantly, the use of filters prior to TE computation for achieving frequency resolution will lead to false positive results due to phase distortions, or will not have the desired frequency-specific effect at all, i.e. TE computed from filtered and unfiltered signals is approximately the same. This latter effect is due to the fact that reducing the power of a signal does not reduce the information contained in it, except for additional effects of signal quantization. Both modes of failure are well known from results on the linear approximations of TE (e.g. via Granger causality, [4–7]).

2. The usual focus on information transfer between a source and a target within a specific narrow frequency band (driven by ideas of synchronization) practically confines the analysis to the linear interaction regime—even when using a nonlinear, model-free, measure like TE. This is because many interesting nonlinear mechanisms of information transfer will actually transform frequencies between source and target.

3. Within-frequency band analyses also ignore the potential many-to-many relationships that source and sender frequencies could have when there is information transfer. For example, a signal at approximately 10 Hz in the source may not seem to transfer information to any specific frequency of the target, yet when considering all frequencies of the raw target signal together, then non-zero TE from 10 Hz at the source to the full signal at the target is observed. In the same way, source signals in two or more bands may have to be considered jointly to reveal TE to the target. On the other hand, multiple bands in the source may carry and transfer identical information to the target, that might be 'double counted' in a naive frequency-resolved analysis. Last, when one frequency is observed sending information and another receiving information, it is not guaranteed that this is actually the same information. In other words, information may be sent from the observed source frequency to all target frequencies jointly while *different* information may be sent from all source frequencies jointly to a specific target frequency.

To circumvent filtering-related problem 1 we here suggest a novel algorithm to obtain frequency resolution of TE without ever filtering the original signals. Instead of filtering the original signals, we apply filtering in the creation of surrogate data representing the null-hypothesis of no information transfer at the frequencies of interest. This way, we use the potential distorting effect of filtering to our advantage, and destroy temporal order in the surrogates instead of changing the power-spectra of the signals. To solve problem 2, we create the frequency-specific surrogate data separately for source- and target-frequencies. This reflects that frequency specific TE is a many-to-many problem, and that within frequency-band analyses may miss most or all of the information transfer. We then discuss problem 3 at the conceptual level, and we explain how splitting of source and target signals into multiple frequencies gives rise to a multivariate problem that is of the 'partial information decomposition' (PID)-type [8, 9]. We use the PID formalism specifically to shed light on the inherent complexity of the problem of spectrally-resolved information transfer in the case where more than one frequency carries information. This also adds an independent motivation to frequency-resolve source and target separately.

## Materials and methods

We start this section with a detailed problem statement, explaining what is and what is not provided by the proposed algorithm. We then also present technical background material on the transfer entropy measure and on the creation of frequency-specific surrogate data in which only a single spectral component has been altered. These two technical sections maybe be skipped at first reading or when not interested in technical details. After this technical background we then present the two core algorithms of this study, which serve to identify source-frequency specific and receiver-frequency specific information transfer.

### Background

**Problem statement and analysis setting.** The aim of the methods proposed here is to determine whether there is statistically significant information transfer orienting from some frequency in an information source, and being received by some—potentially different—

frequency in a target of the information transfer. We pose this question in a network setting. This means that we are interested in the above frequency-resolved information transfer conditional on the other processes in the network. For the algorithms presented here we explicitly assume that the multivariate network identification problem has been solved, i.e. that the information transfer between a source and a target, conditional on the relevant rest of the network, is genuine. By this we mean that the information flows directly from source to target, and does not flow via any intermediate node that we have data from. We also assume that the information transfer in question is not an apparent information transfer due to common driver effects (see e.g. [10]). This setting can be achieved by computing either multivariate transfer entropies directly (see next section), possibly via some greedy approximation [11, 12], or via a computation of bi-variate transfer entropies in combination with another approximate correction method [10].

Specifically, we assume a network of $M + 1$ nodes. For any spectrally resolved analysis these are conceived of as $M$ potential source processes $\mathcal{S}_i, \quad i = 1 \ldots M$, and 1 potential target process $\mathcal{T}$. These stochastic processes are represented by multidimensional state vectors $\mathbf{S}_i, \mathbf{T}$ (see e.g. [13]), covering the relevant past of the processes. This relevant past can be obtained via the methods described in [14] and [11]. The assignment of a process to be the target can be repeated until the whole network has been covered. Ultimately, we obtain a set of targets with their respective information sources in space and time. For each information source of a target we have in the process also computed the frequency-independent information transfer from source to target—conditional on the other sources in the form of a conditional, or multivariate, transfer entropy.

Last, we would like to stress again that in this setting problems related to cascade and common driver effects in the network can be considered as solved, or solved to the degree possible by non-interventional methods. Our methods to compute *spectrally-resolved* information transfer can then be seen as post-processing step aimed at providing the more fine grained spectrally-resolved perspective, i.e. the methods presented here are not aimed at providing network identification.

**Technical background: Transfer entropy and pre-computation of multivariate transfer entropy.** Transfer entropy (TE) as the fundamental measure of information transfer was introduced first in [1] in a bivariate framework for two random processes $\mathcal{X}, \mathcal{Y}$ as:

$$TE(\mathcal{X} \rightarrow \mathcal{Y}) \coloneqq I(Y^+ : \mathbf{X}^- | \mathbf{Y}^-) \tag{1}$$

where $I(\cdot : \cdot | \cdot)$ is the conditional mutual information and $Y^+, \mathbf{Y}^-, \mathbf{X}^-$ are, respectively, a future random variable of the process $\mathcal{Y}$, a vector of suitably chosen past random variables of the past of that process, and a suitably chosen vector of past random variables of the process $\mathcal{X}$ (see [1, 11, 13–15] for considerations on the correct choice of the past random variables). TE measures the amount of information transferred between a single source and a single target process. In a network setting where multiple sources may interact to transfer information to a target, or where multiple sources transfer information redundantly to a target, TE needs to be extended to a multivariate formalism in order to avoid spurious results. Thus, we need to measure the information transfer from a single source to a target, but now in the context of all other relevant sources in an observed network [10, 16]. In other words, the multivariate TE (mTE) for a system of $M$ random source processes $\mathcal{S}_1, \ldots, \mathcal{S}_M$, and a target $\mathcal{T}$, observed over $D$ discrete time steps, represented by random vectors $\mathbf{S}_i = (S_{i,1}, \ldots, S_{i,D})$, measures the information transfer as a *conditional* mutual information of the following form:

$$mTE_{\text{tot}}(\mathcal{S}_i \rightarrow \mathcal{T} | \mathbf{S}_{<t}^{\backslash \{\mathbf{S}_i, \mathbf{T}\}}) = I(T_t : \mathbf{S}_{i,<t} | \mathbf{T}_{<t}, \mathbf{S}_{<t}^{\backslash \{\mathbf{S}_i, \mathbf{T}\}}) , \tag{2}$$

where $\mathcal{T}$ is a process considered as the current target of the information transfer, $\mathbf{S}_{i,<t} = (S_{i,t-\delta}, \ldots, S_{i,t-k})$, with $\delta \leq k$ is a vector of random variables chosen from $\mathbf{S}_i$ from the past of the current time point $t$. The last element $k$ of this vector is chosen such that $\mathbf{S}_{i,<t}$ renders $T_t$ conditionally independent of all variables in the process $\mathbf{S}_i$ that are further back in time. The delay parameter $\delta$ is chosen such that it reflects the physical delay in the system (see [14]). $\mathbf{S}_{<t}^{\backslash\{\mathbf{S}_i,\mathbf{T}\}}$ signifies the collection of past random variables from all other processes except $\mathbf{S}_i$ and $T$. Last, the subscript 'tot' means that we are focusing on the total information transferred from all relevant past variables of the process $\mathbf{S}_i$, rather than the contribution of each individual variable $S_{i,d}$.

Computing the $mTE_{tot}$ in Eq 2 exactly is an NP-hard problem [17], and approximations are necessary for practical use. This problem was recently addressed in [12, 18], with the implementation of an approximate greedy algorithm in the IDT$^{xl}$ toolbox, which allows a large-scale directed network inference with mTE [11] and is freely available from GitHub (https://github.com/pwollstadt/IDTxl). IDTxl performs a greedy algorithm with an iterative sequence of statistical steps to infer the 'relevant' sources of the network, thus reducing the dimensionality of the problem, and allows to properly construct the nonuniform embedding of source and target time-series [19, 20], i.e. it also yields approximations for parameters like $\delta$ and $k$. We note that even using the greedy approximation, computations of $mTE_{tot}$ scale as $n^3$, with $n$ being the number of processes under consideration. Last we note that any interpretation of empirical values for $mTE_{tot}$ from real-world systems needs to take into account also problems of partial observability (not all relevant nodes of the system are recorded) [21] and coarse graining (recordings from 'nodes' of the system represent summary signals from more microscopic parts).

The $mTE_{tot}$ estimated from the original data will be the test statistic of interest for our spectral mTE algorithm in which it is statistically tested against a non-parametric null distribution of the same measure computed from frequency-specific surrogate data constructed by the maximum overlap discrete wavelet transform (MODWT). These surrogate data represent the null hypothesis of no information transfer being sent (or received) by the frequency of interest.

In the exposition of the spectral mTE algorithm below we will assume that multivariate network reconstruction, and $mTE_{tot}$ estimation for the original data have been performed and a set of sources $\mathbf{S}_{i,<t}$ significantly contributing $mTE$ for the target $\mathcal{T}$ has been identified. This set of significant source processes with respect to a target will be passed to the spectral TE algorithm, to identify TE relations at specific frequencies in these sources and the target. The algorithm is applied to find the spectral components of contributing to the overall information transfer one source at a time $\mathbf{S}_{i,<t}$, but is of course repeated over all relevant sources of a target.

All values of $mTE_{tot}$ were estimated using the nearest-neighbour based methods proposed by Kraskov (estimator 1 in [22] as implemented in the IDT$^{xl}$ toolbox [12]. For all simulations we reported the sample size of data points used. For the empirical data see descriptions of the data in the corresponding sections below.

**Technical background: Maximum overlap discrete wavelet transform.** Several methods have been established for surrogate data creation, each with its own limitations and advantages (see [23] for a review). Among many, wavelet-based methods allow to create frequency-specific surrogate data through randomization of the wavelet coefficients [24]. In particular, wavelet-based surrogates that preserve the local mean and the variance of the data were introduced by [25]. Similarly to [26], we employ the Maximal Overlap Discrete Wavelet Transform (MODWT), to transform the data in the wavelet domain. The MODWT is well defined for

time-series of any sample size and produces wavelet coefficients and spectra unaffected by the transformation. [26].

The MODWT of a time-series $X = (X_0, \ldots, X_{N-1})$ of $J_0$ levels, where $J_0$ is a positive integer, consists of $J_0 + 1$ vectors: $J_0$ vectors of wavelet coefficients $\tilde{\mathbf{W}}_1, \ldots, \tilde{\mathbf{W}}_{J_0}$ and an additional vector $\tilde{\mathbf{V}}_{J_0}$ of scaling coefficients, all with dimension $N$ (our exposition of the MODWT closely follows that of [27], pages 159-205). The coefficients of $\tilde{\mathbf{W}}_j$ and $\tilde{\mathbf{V}}_{J_0}$ are obtained by filtering $X$, namely:

$$\tilde{W}_{j,t} = \sum_{l=0}^{L_j-1} \tilde{h}_{j,l} X_{t-l \quad \mod N},$$ (3)

$$\tilde{V}_{j,t} = \sum_{l=0}^{L_j-1} \tilde{g}_{j,l} X_{t-l \quad \mod N},$$ (4)

where $\{\tilde{h}_{j,l}\}$ and $\{\tilde{g}_{j,l}\}$ are the $j$th level MODWT wavelet and scaling filter, with $l = 1, \ldots, L$ being the length on the filter and $L_j = (2^j - 1)(L - 1) + 1$. We can write the above in matrix notation as:

$$\tilde{\mathbf{W}}_j = \tilde{\mathcal{W}}_j X$$ (5)

$$\tilde{\mathbf{V}}_{J_0} = \tilde{\mathcal{V}}_{J_0} X$$ (6)

where each row of the $N \times N$ matrix of $\tilde{\mathcal{W}}_j$ has values denoted by $\{\tilde{h}_{j,l}^{\circ}\}$, while $\tilde{\mathcal{V}}_j$ has values denoted by $\{\tilde{g}_{j,l}^{\circ}\}$, where $\{\tilde{h}_{j,l}^{\circ}\}$ and $\{\tilde{g}_{j,l}^{\circ}\}$ are the periodization of $\{\tilde{h}_{j,l}\}$ and $\{\tilde{g}_{j,l}\}$ to circular filter of length $N$ [27]. Thus, the MODWT treats $X$ as if it were periodic, such periodic extension is known as 'circular boundary condition' [27]. Finally, the time series $X$ can be retrieved from its MODWT with [27]:

$$X = \sum_{j=1}^{J_0} \tilde{\mathcal{W}}_j^T \tilde{\mathbf{W}}_j + \tilde{\mathcal{V}}_{J_0}^T \tilde{\mathbf{V}}_{J_0}$$ (7)

While, the coefficients $\tilde{\mathbf{V}}_{J_0}$ represent the unresolved scale [26, 27], and capture the long term dynamics of $X$, the coefficients $\tilde{\mathbf{W}}_j$ are associated with changes of the underlying dynamics, at a certain scale, over time. If $N = 2^J$ and we set $J_0 = J$, then a full decomposition is performed and the scale $\tilde{\mathbf{V}}_{J_0}$ retains only the average constant of the data with all other information represented in the wavelet coefficients [26, 28]. Since in many applications a full decomposition is not necessary (e.g. the dynamic of a physical system is meaningful over a certain frequency range only), $J_0$ can be set to any integer $J \leq \lfloor (\log_2(N)) \rfloor$ so that the decomposition at any scale is shorter than the total length of the time series [29]. The selection of $J_0$ determines the number of scales of resolution with the MODWT coefficients at a certain scale $j$ related to the nominal frequency band $|f| \in (1/2^{j+1}, 1/2^j)$ [27]. Moreover, given $\tilde{\mathbf{W}}_j$ and $\tilde{\mathbf{V}}_j$ it is possible to reconstruct the time-series $X$ through the inverse MODWT (IMODWT). If the coefficients are not modified, the IMODWT returns the original time-series $X$ [27].

## Algorithms

**Algorithm I: Identifying source- or receiver-frequency specific TE.**

**Core idea.** The core idea of the proposed algorithm is to never apply any frequency-specific signal processing to the original data from which TE is computed, as this is known to come with a whole host of problems [5, 7]. Rather, frequency-specificity is obtained by destroying TE-relevant signal properties (like temporal order) in a frequency-specific manner *in the surrogate data* and to then look for a significant drop in mTE in these surrogate data compared to the original mTE via non-parametric statistical testing. To this end, we create surrogate data via an invertible wavelet transform (maximum overlap discrete wavelet transform, MODWT) and a frequency (scale-) specific scrambling of the wavelet coefficients in time. Thus, in the surrogate data temporal order and phase relations are destroyed specifically in the band of interest, while the power spectra of the signals are preserved. The null hypothesis embodied by the surrogate data thus that the wavelet-coefficients of the frequency-component of interest in the source time series are exchangeable in their temporal relation to the target time series with respect to the information transferred between source and target.

As frequency separation is never perfect, we confine ourselves in most cases to only interpreting the wavelet-scale or frequency where the difference between the median of the distribution of mTE from surrogate signals and the value of total original multivariate TE ($mTE_{tot}$, see next) is largest. If multiple, well separated maxima of this difference can be observed, all of them may be interpreted. However, Bonferroni-correction for multiple testing should be applied in this case.

**Implementation for source-frequency specific information transfer.** As introduced above, we obtain a measure of frequency-specific information transfer by creating surrogate datasets in which the temporal ordering of the signals has been destroyed for specific spectral components of these signals—by first transforming into the frequency domain, then scrambling wavelet coefficients for a specific frequency and last transforming back to the time domain to obtain a surrogate dataset. Naively one may be tempted to apply this process to source and target processes at the same time. Yet, this approach would limit the analysis to within-band effects. As laid out in the introduction and also detailed in section *Frequency resolved TE as a partial information decomposition problem*, this would ignore the multivariate nature of the problem. Therefore, we apply the creation of frequency-specific surrogate data separately to source and target processes, i.e. we apply two variants of the analysis—one measuring source-frequency specific information transfer and the other measuring target-frequency specific information transfer. A combination of the results of both analyses is sometimes possible when carefully considering before the multivariate nature of the problem and prior knowledge (see section *Relation of the partial-information decomposition framework and the SOSO-algorithm* below).

We will now detail the algorithm variant for the measurement of source-frequency specific information transfer and report the relevant differences for measuring target-frequency specific information transfer afterwards.

Measuring source-frequency specific mTE relies on five main steps:

1. Perform a wavelet decomposition of the source time series through the MODWT to obtain a time-frequency representation of $\mathbf{S}_i$ in $J_0$ scales.

2. At the j[th] scale of the MODWT decomposition shuffle the wavelet coefficients to destroy information carried by the scale (frequency band)

3. Apply the inverse wavelet transform, IMODWT, to get back the time representation of the time series

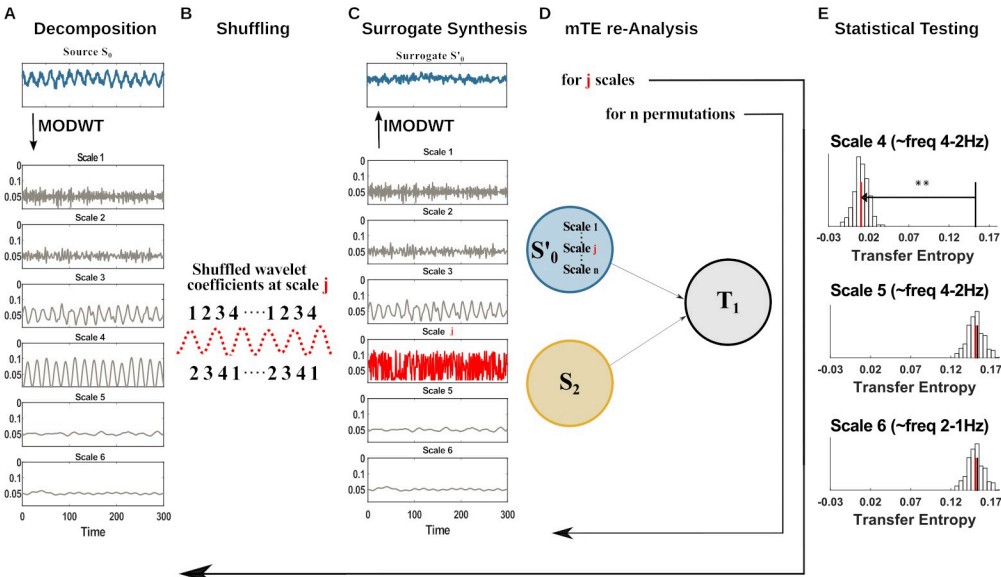

**Fig 1. Spectral TE algorithm pipeline.** (A) The neural signal (blue) is converted to a time-frequency representation (grey) using the invertible maximum overlap discrete wavelet transform (MODWT). (B) At a frequency (wavelet scale) of interest in the source (or the target) the wavelet coefficients are shuffled in time, destroying its connection to the target (or source). (C) The signal is recreated by the inverse MODWT. (D) The transfer entropy for the original and many shuffled signals is computed. (E) A statistical test determines whether the shuffling reduced the information transfer, indicating that the transferred information was indeed encoded at the specific frequency. Each panel here shows the distribution of $mTE'$ values (vertical bars) obtained from surrogate data where the wavelet coefficients of the scale of interest were shuffled, the median of this distribution (red line), and the original transfer entropy (black line). The analysis and the testing is repeated for all scales of interest (here 4, 5, 6).

4. Compute the $mTE'$ between the surrogate source and the target, conditional on all other significant sources in the network.

   a. Repeat step 2 to 4 for a number of permutations to build a surrogate data distribution.

   b. Repeat step 1 to 4 for all $J_0$ scales.

5. Test whether the original $mTE_{tot}$ is above the $1 - \frac{\alpha}{J_0}$ quantile of the surrogate-based distribution of $mTE'$ values at each scale, i.e. perform a significance test with respect to the surrogate-derived distribution.

The operations implemented in the five steps are illustrated in Fig 1 and described in detail hereafter.

Step 1. The source time-series is decomposed once into $J_0$ scales through the MODWT (Fig 1A). As introduced in section *Maximum Overlap Discrete Wavelet Transform* this decomposition gives a set of details coefficients $\tilde{\mathbf{W}}_{J_0}$ and an additional set of approximation coefficients $\tilde{\mathbf{V}}_{J_0}$. The latter is saved in this first step and utilized only in step 3, without any modification. Only the $\tilde{\mathbf{W}}_{J_0}$ coefficients at the $j^{th}$ scale under analysis are subjected to step 2. The current implementation uses a Least Asymmetric Wavelet (LA) as mother wavelet of length 8 or 16, since both lengths showed to be robust against spectral leakage and do not relevantly suffer from boundary-coefficient limitations. [25, 27, 30].

The creation of surrogate data for subsequent statistical testing comprises of the following steps 2 and 3.

Step 2. The frequency-specific information transfer between source and target is destroyed by shuffling the $\tilde{\mathbf{W}}_{J_0}$ wavelet coefficients one scale at a time. The $j^{th}$ scale under analysis is shuffled by randomly permuting the coefficients $\tilde{\mathbf{W}}_j$, whereas all the other scales decomposed by the MODWT stay intact (Fig 1B, $j^{th}$ scale in red). We implement two alternative methods for the creation of surrogate data: a Block permutation of the wavelet coefficients [24] and the Iterative Amplitude Adjustment Fourier Transform (IAAFT) [24, 26]. Since there is no unique method of surrogate data creation and in many cases the employment of one method or another much depends on the specific analysis carried out by the user, we describe the two methods and the input parameters in section *Resampling methods and the free parameters*.

Step 3. The unchanged set of coefficients, $\tilde{\mathbf{W}}_{J_0 \setminus j}$, the unchanged $\tilde{\mathbf{V}}_{J_0}$'s, and the permuted coefficients at scale $j$ ($\tilde{\mathbf{W}}_j$) are submitted to the IMODWT, to reconstruct the surrogate source signal, $\mathbf{S}'_i$, in the time-domain (Fig 1C). This step is identical for both of the implemented surrogate-data creation methods: Block permutation of the wavelet coefficients and IAAFT. The reconstructed source $\mathbf{S}'_i$ (*source surrogate*) differs from the source $\mathbf{S}_i$ only on the shuffled $j^{th}$ scale. In this way, we destroy the source-target information transfer only if the information transfer is carried by the $j^{th}$ scale, otherwise the information transfer stays the same.

Step 4. With $\mathbf{S}'_i$ we compute again the $mTE_{tot}$ on the network previously identified. We illustrated this step in Fig 1D. Let $\mathbf{S}_{i,<t}$ be the set of past variables of the *selected sources* and $\mathbf{T}_{<t}$, the past variables of the *selected target* previously found in the network analysis, with $\mathbf{S}'_{i,n}$ being the $n$-th *source surrogate* under analysis in the network at scale $j$; then, the $mTE'$ for the surrogate data is:

$$mTE' = mTE(\mathbf{S}'_{i,n} \rightarrow \mathbf{T} | \mathbf{S}^{\setminus \{\mathbf{S}'_{i,n}, \mathbf{T}\}}_{<t}) = I(T_t : \mathbf{S}'_{i,n,<t} | \mathbf{T}_{<t}, \mathbf{S}^{\setminus \{\mathbf{S}'_{i,n}, \mathbf{T}\}}_{<t}) \quad (8)$$

The algorithm is repeated from step 2 to step 4 for *n permutations*, with $n = 1, \ldots, N$, to create a distribution of surrogate $mTE'_n$ values; $N$ is set according to the desired critical level for statistical significance (including Bonferroni correction for the number of scales, see below). Subsequently, all the $J_0$ scales decomposed by the MODWT in step 1 are subjected to step 2, step 3 and step 4, such that $J_0$ separate distributions of $mTE'_n$-values, one for each scale, are obtained.

Step 5. As a final step, the $mTE_{tot}$ is tested for statistical significance against the $J_0$ different distributions of $mTE'$ surrogate values. If the $\mathbf{S}^j_i$ (where $j$ is one of the scales decomposed by the MODWT) carries any information transfer to the target $T$, a significant drop of the $mTE'$ surrogates will be observed. This step is applied for all $J_0$ scales under analysis and a Bonferroni correction is applied such that each individual scale is tested at the significance level $\alpha/J_0$.

Additionally, each scale analyzed is plotted, see Fig 1E, and we restrict ourselves to interpret only the scale that shows maximal distance (or well separated local maxima) from the original $mTE_{tot}$, $\max_j(mTE_{tot} - \tilde{mTE}')$, where $\tilde{mTE}'$ denotes the median of the surrogates distribution. We consider the maximal distance in addition to the statistical significance test because frequency decomposition is never perfect (e.g.

leakage, noise and wavelet bands overlap). Indeed, validation of the algorithm on synthetic data (section 3) shows that the maximal distance reliably reflects the ground truth in the sender-receiver frequency information transfer, independently of the method employed for surrogate construction, whereas the statistical significance test can suffer from leakage effects on adjacent scales. Obviously, this limits the detectability of frequency-specific $mTE_{tot}$ to one source frequency and may be overly conservative. Thus, in scenarios, where information transfer from multiple sources is strongly expected *a priori*, or where the length of the data allows for vanishing leakage effects, the above restriction may be lifted.

**Implementation for target-frequency specific information transfer.** To measure the target-frequency specific *mTE*, we apply the same algorithm as before, but this time we create frequency-specific surrogate data from the target time series:

1. Perform a wavelet decomposition through the MODWT to obtain a time-frequency representation of the target time series' present state, $T_t$, the target of the multivariate information transfer from $\mathbf{S}_i$ to $\mathbf{T}$.

2. At the $j^{th}$ scale of the MODWT decomposition shuffle the wavelet coefficients to destroy information entering the scale in the target or amplitude-phase relations. This step is different from the shuffling in the source algorithm implementation; here, we destroy only the target current value $T_t$ to obtain $T'_{t,n}$, where $T'_{t,n}$ is the $n$-th *target surrogate* under analysis in the network at scale $j$ and leaving the target past set, $\mathbf{T}_{<t}$, intact,

$$mTE' = mTE(\mathbf{S}_i \to \mathbf{T}'_n | \mathbf{S}^{\setminus\{\mathbf{S}_i,\mathbf{T}_n\}}_{<t}) = I(T'_{t,n} : \mathbf{S}_{i,<t} | \mathbf{T}_{<t}, \mathbf{S}^{\setminus\{\mathbf{S}_i,\mathbf{T}_n\}}_{<t}) \tag{9}$$

3. Apply the inverse wavelet transform, IMODWT, to reconstruct the time series in the time domain.

4. Compute the $mTE'$ between the source and the *target surrogate*, conditional on all other significant sources in the network.

   a. Repeat step 2 to 4 for $N$ permutations to build a surrogate data distribution.

   b. Repeat step 1 to 4 for all $J_0$ scales.

5. Check for which scale, $j$, the difference between the original $mTE_{tot}$ and the median of the $mTE'$ distribution is maximal, and determine statistical significance for this scale, similar to the source-frequency implementation.

**Algorithm II: Testing for direct information transfer from source to receiver frequencies.** Consider the following scenario where a certain frequency in the source transfers information to a certain frequency in the target (Fig 2A). We would like to then identify these two related frequencies in the source and the target and to determine that there is indeed transfer information *between them*—and not to other, more broadband parts of the spectrum. In other words, we want to exclude the possibility that the source frequency sends information to many other frequencies in the target, potentially even missing the identified target frequency, while the identified target frequency receives information from many source frequencies, potentially excluding the identified source frequency (Fig 2B)—such that the direct information transfer between the two identified frequencies is actually absent. We also want to exclude the

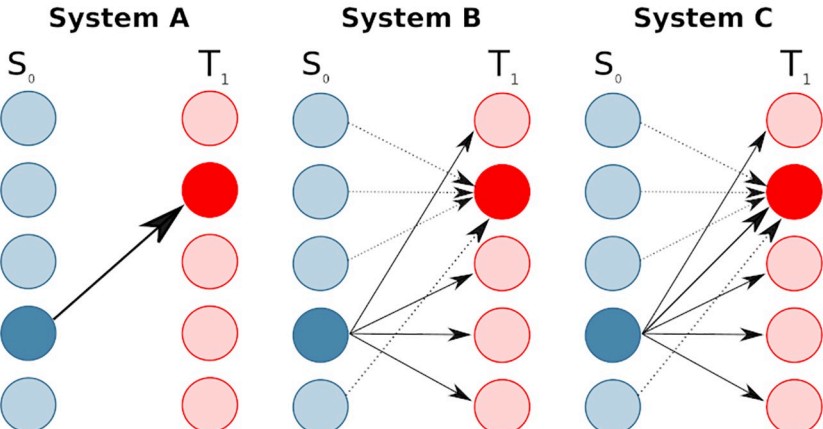

**Fig 2. Three systems with the same identified sending and receiving frequencies (indicated by the darker blue and red colors), but a different structure of information transfer.** In system A one source and one target frequency take part in a direct transfer of information between them. In system B one source frequency sends information to all target frequencies except the identified target frequency. This one target frequency, in turn, receives other information from all source frequencies except the identified source frequency. In system C the same source frequency sends information redundantly into all target frequencies, while one target frequency receives (partially different) information redundantly from all source frequencies.

possibility that the direct information transfer between source and target is entirely redundant with other spectral components of information transfer (Fig 2C).

When applying algorithm I to the target in the setting assumed above (Fig 2A) we will observe a drop in mTE for the surrogate data at the source frequency driving the information transfer, and the target frequency receiving it. If we applied the same algorithm to data where the phase of the sending frequency had been destroyed *beforehand*, then no information transfer should be seen from the source, and thus, also algorithm I applied to the target should also not yield a drop anymore (for the surrogate data with an additionally scrambled target, see also Fig 3B and 3C). Since in this procedure we first swap out the source frequency and then the target frequency in addition, we also refer to algorithm II as the 'swap-out swap-out (SOSO)' algorithm from here on. This SOSO algorithm is to be applied after algorithm I has identified specific source and specific target frequencies. That is, we apply the SOSO algorithm as a *post-hoc* analysis.

**Algorithm implementation.** In the following we describe a version of the implementation of the SOSO algorithm, in which we first destroy the target $\mathbf{T}^r$ and subsequently also the source time-series $\mathbf{S}_i^j$, where $j$, $r$ are the scales of interest (Fig 3). The algorithm can also be applied in the opposite direction by first destroying the source $\mathbf{S}_i^j$ and subsequently the target $\mathbf{T}^r$.

First, let $\delta_{TE_{S_i}}^j = mTE_{tot} - m\tilde{T}E'$, be the distance between the $mTE_{tot}$ and the $m\tilde{T}E'$, computed with Algorithm I at scale $j$ from source $\mathbf{S}_i$ and target $\mathbf{T}$. Then, Algorithm II comprises two main steps:

1. For $N$-times, scramble the target time-series current value $\mathbf{T}_t$ at scale $r$ with one of the implemented shuffling methods and compute the $mTE'_{T,n}$ (as described in the subsection *Implementation for target-frequency specific information transfer*). Here the subscript 'T' indicates that the target has been shuffled.

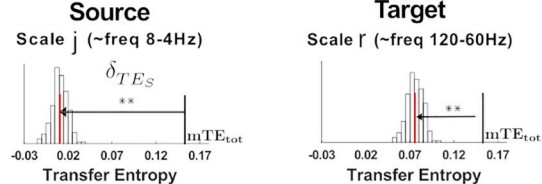

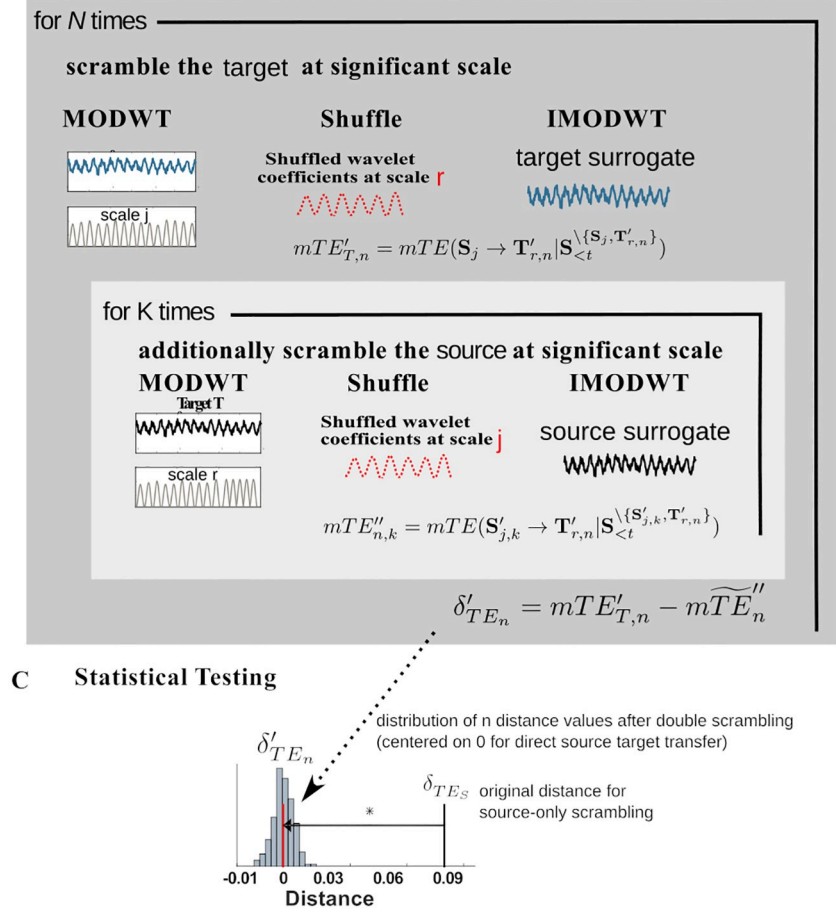

**Fig 3. Algorithm II (SOSO).** Algorithm to determine whether information transfer exists from an identified information source scale to an identified target scale. (A) Results from the initial analysis using Algorithm I indicating significant information transfer emanating from one scale (source scale $j$) and significant information reception at a target scale (target scale $r$). (B) To test if the information send from the source scale is indeed the information that is received at the target scale do the following: scramble the target at the relevant scale $N$ times and note the $mTE'_{T,n}$ values. For each such scrambled target then apply algorithm I for the source, i.e. scramble the relevant source scale $K$ times and note the distribution of the $mTE''_{n,k}$ values. Compute the drop in $mTE$ obtained for the $n-th$ target shuffling with respect to the median $m\tilde{T}E''_n$ of the distribution of source-and-target shuffled $mTE''_{n,k}$ values, $\delta'_{TE_n}$. (C) Statistically test the original target drop $\delta_{TE_S}$ against the distribution of the $\delta'_{TE_n}$. A significantly larger value of $\delta_{TE_S}$ indicates that information send by the source scale is indeed received by the target scale.

a. For each permutation, create an inner loop running $K$-times, where also the source $S_i$ is destroyed at scale $j$, as before, and compute $mTE''$, to obtain a distribution of $mTE''_{n,k}$ values. Here the double prime symbol signifies that both the target and the source scale of interest have been destroyed.

 b. Compute the distance between $mTE'_{T,n}$ and the median over $k$, $m\tilde{T}E''_n$ of the distribution $mTE''_{n,k}$, to obtain a distribution of distances $\delta'_{TE_n} = mTE'_{T,n} - m\tilde{T}E''_{n,k}$

2. Check whether $\delta^j_{TE_{S_i}}$ is at the extreme upper end of the distribution of surrogate distances $\delta'_{TE_n}$.

The SOSO algorithm performs $(N+1) * K$ mTE computations at the selected source scale and target scale pairs, where $N$ and $K$ are the number of permutations. Although the SOSO algorithm could be, in principle, applied to all possible source- and target-scale combinations of the identified network, we discourage this approach as pointed out in section *Advantages and drawbacks of the proposed methods*.

## Resampling methods and the free parameters

In this section we provide a description of the resampling methods implemented in the spectral mTE algorithms above and of their free parameters. These methods are used to shuffle the wavelet coefficients for the creation of the surrogate data.

**Resampling block size.** The block resampling technique has been used extensively in surrogate data generation (see for example [24]). In this paper, we consider a resampling block size of 1, which can be thought of as a simple random permutation of the wavelet coefficients. The block size is an input parameter of the spectral TE and it can be set by the user (e.g. the block size is set to 32 in [24]).

**Iterative Amplitude Adjustment Fourier Transform.** The IAAFT method relies mainly on the work of [26], where a detailed description of the implementation and different applications can be found. We used the same algorithm with two fundamental changes. First, we apply the IAAFT method at one scale at time. This is motivated by the necessity to destroy putative TE information one scale at a time, keeping the contribution of the other scales intact. Second, with the IAAFT we do not apply any threshold to retain wavelet coefficients intact at a certain scale (to refer to [26] we set the threshold $p = 0$, so all coefficients are randomly shuffled and go to the iterative amplitude adjustment) since our goal was not to have a qualitative analysis between surrogate data and original data.

**Choice of target history coverage.** Eq 2, in *Technical background: Transfer entropy and pre-computation of multivariate transfer entropy*, contains the candidate source past set $S_{<t}$ and the candidate target past set $T_{<t}$ which have to be defined to compute $mTE$. In the presented simulations, we set the maximum lag of the target to cover at least 1/4 of the cycle of the lowest frequency of interest (e.g. if the lowest frequency was 4 Hz, we covered 1/4 of the cycle of 4 Hz). The maximum source lag was set to 3 samples lag, since the true delays were known in the simulations (1 or 2 samples lag). In case of other applications the maximum source lag should span a plausible number of samples for the system under study (e.g. a range of plausible axonal conduction delays in the case of neural data). We note that if sufficient computational resources are available, then it is possible to set a very generous limit on the covered history of the target and let the iterative algorithms of IDT$^{xl}$ decide where to truncate the target history.

**A cautionary note on frequency specific information transfer versus cross frequency coupling.** Before concluding, we would like to stress that our novel algorithm should not be misunderstood as an analysis technique to estimate cross-frequency coupling (CFC, see [31] and references therein). This is because, first, information transfer and coupling are conceptually different (and it is transfer that is more important when trying to understand a computation, whereas coupling is important to understand the biophysics and dynamics of a system, [32, 33]). Ironically, however, most of the methods cited in the field of cross-frequency

*coupling* do not yield strong evidence of physical coupling, but remain correlation-based (see discussion in [31]). Thus, these methods can be seen as some form of coarse approximation to cross-frequency information transfer. Yet it should be kept in mind that these methods lack directionality, and do not quantify information proper, but some other measure of statistical dependency (often linear). Also, these methods typically focus on specific dependencies between the phase-evolution and the amplitude envelope of the recorded signals (see [31], and references therein). In contrast, a full information-theoretic analysis takes all of these into account simultaneously.

We would also like to stress here again that the concept of information being transferred from individual source frequencies to individual target frequencies—as it is expressed in the specific wording 'cross-frequency'—does not reflect the actual complexity of the problem.

## Results

To test the ability of the proposed spectral mTE algorithm to successfully estimate frequency-specific sender-receiver information transfer, we employed multiple synthetic simulations, where the information transfer was known (ground truth). Additionally, we demonstrate the application of algrithms I and II to two neural data sets. The first is a human neuroimaging dataset, acquired with Magnetoencephalography (MEG); the second dataset consists of local field potential (LFP) recordings from the ferret cortex. The simulations for individual scenarios and details of the neural MEG and LFP data are described below. All analysis were performed with a block permutation of the wavelet coefficients method (to construct surrogates) and LA(8) as mother wavelet, if not stated otherwise.

### Example 1: Uncoupled system, no information transfer

At first, we tested the behaviour of the spectral mTE algorithm in the case of an uncoupled system. Since no information transfer occurs (at any frequency band), the applications of the spectral mTE algorithms should not reveal any significant drop in term of TE, at any scale.

We simulated two uncoupled signals with the following equations:

$$S_0(t) = A * cos(2\pi f_1 t + \theta) + w_1 \tag{10}$$

$$T_1(t) = A * cos(2\pi f_2 t + \theta) + w_2 \tag{11}$$

Here $A$ is the amplitude of the signal and is set to 1 if not stated otherwise, $\theta$ is a uniform random variable between 0 and $2\pi$, $f_1 = 5$ Hz, $f_2 = 50$ Hz and $w_1$, $w_2$ are samples of i.i.d Gaussian noise process with a standard deviation of 1. We simulated 10 seconds and 100 trials with a sampling rate of 125 Hz (125000 samples).

The computation of $mTE$ without spectral resolution, correctly, did not show any significant TE in either direction. Nevertheless, just for demonstration purposes, we applied the spectral TE algorithm I, considering $S_0$ as source and $T_1$, as target (this choice is arbitrary since there is no real coupling). The results showed no significant drop at any scale for both source and target, as expected (Fig 4B). Additionally we ran the spectral mTE analysis 500 times to estimate if the alpha level chosen (0.05/ number of scales at source or target), reliably protects from false positive results. At each scale, for source and target, no false positives, with $\alpha/J_0$ (where $J_0$ is set to 5 in this simulation), were found, indicating that our measure is in fact somewhat conservative here.

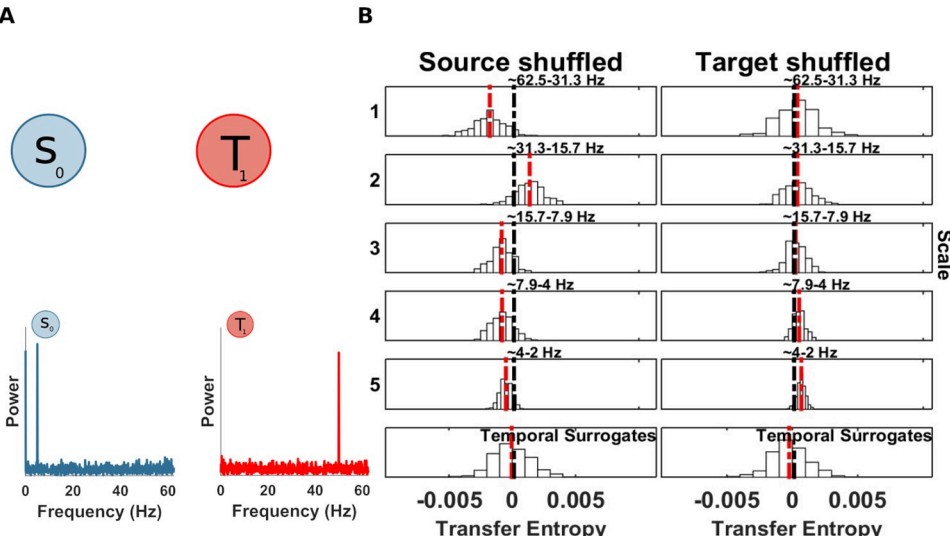

**Fig 4. Spectrally resolved transfer entropy for the null-case (example 1).** (A) Top, a 'source' $S_0$ and a 'target' $T_1$ of an uncoupled system. Bottom, power spectra of $S_0$ and $T_1$. (B) Spectrally resolved Transfer Entropy. Each panel, except those at the bottom, shows the $mTE'$ distribution obtained from the surrogate datasets with shuffled coefficients at the scale indicated to the left, or, equivalently, the frequency band indicated at the top of each panel. White bars represent histograms of surrogate data, i.e. relative frequencies in (a.u.), the red dashed line is the median of the surrogate $mTE'$ distribution, the black dashed line is the original $mTE$ value. The horizontal black line indicates the distance $\delta_{TE}$ between the original $mTE$ and the median of the surrogate distribution (\*\*, $p < 0.005$; \*, $p < 0.05$). These display conventions will be kept for figures displaying spectrally resolved TE analyses. The temporal surrogate analysis using surrogates constructed by permuting blocks of samples in the time-domain is shown in the bottom row. No significant drop of the shuffled wavelet coefficients could be found, since no information transfer occurred between a putative source and the target site. (Note that the choice of source or target here is arbitrary since no coupling was simulated).

## Example 2: Information transfer from one source to one target frequency in a bivariate system

Next, we simulated a simple bivariate scenario, where a single source $S_0$ is (multiplicatively) coupled to a target $T_1$ that oscillates at a much faster frequency, such that the amplitude of the target is modulated by the phase of the source, leading to a cross-frequency information transfer (CFIT) (Fig 5A). Moreover, the source is coupled to the target with a delay of 2 samples. The synthetic data are generated according to the following equations:

$$S_0(t) = A * cos(2\pi f_1 t + \theta) + w_1 \tag{12}$$

$$T_1(t) = A * cos(2\pi f_2 t + \theta) * S_0(t - 2) + w_2 \tag{13}$$

Where, $f_1 = 6$ Hz, $f_2 = 50$ Hz and $w_1, w_2$ are samples of i.i.d Gaussian noise process with a standard deviation of 1. We simulated 10 seconds and 100 trials with a sampling rate of 125 Hz (125000 samples).

First, we performed a TE analysis to recover the source-target information transfer. Table 1 reports the result of the TE analysis. We recovered the true direction of interaction from $S_0$ to $T_1$, with a maximal TE at a lag of 2 samples (16 ms), as simulated. Then, we applied the spectral mTE algorithm to the identified source-target relation to recover the sender- and receiver-frequency information transfer. We found a significant drop of $mTE^*$ in the source $S_0$ at source-scale 4 (frequency band $4 - 7$ Hz), as expected (Fig 5B). The amplitude-phase modulation of the target $T_1$ is visible at target scale 1 (frequency band $62 - 31$ Hz)—again as expected. In this relatively simple scenario in terms of sender-receiver frequency relation, the spectral mTE is

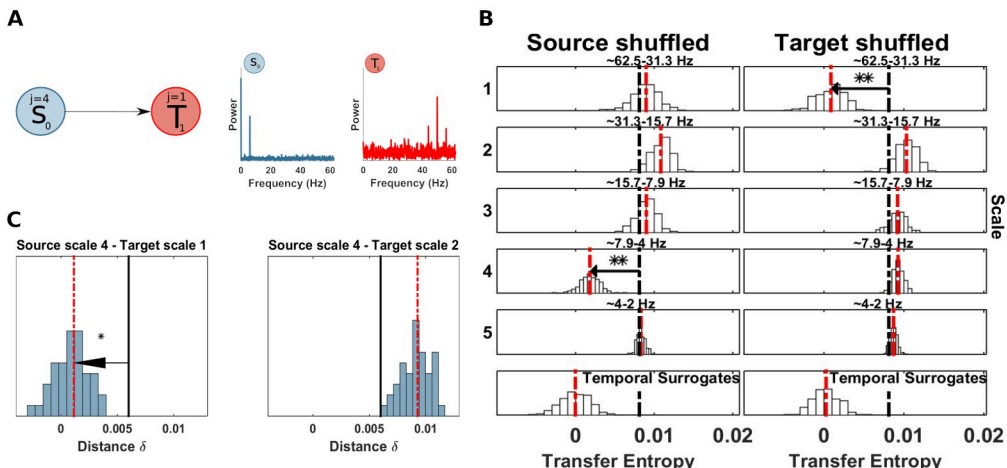

**Fig 5. Spectrally resolved transfer entropy for example 2.** (A) Left, a source $S_0$ is unidirectionally coupled, at scale $j = 4$ (frequency band $4 - 7$ Hz), with a target $T_1$ at scale $j = 1$ (frequency band $31 - 62$ Hz). Right, power spectra of $S_0$ and $T_1$. (B) Spectrally resolved Transfer Entropy. See Fig 4 for display conventions. Information transfer correctly drops when wavelet coefficients are selectively shuffled at scale 4 at the source ($S_0$, left column). The corresponding reception of information at the target ($T_1$) is shown on the right, where a drop for shuffled wavelet coefficients is observed for the frequency band receiving the information in this simulation (i.e. scale 1). The temporal surrogate analysis using surrogates constructed by permuting blocks of samples in the time-domain is shown in the bottom row. (C) SOSO analysis. Blue bars display the distribution of distances $\delta'_{TE}$ between the median of the surrogate data distribution with a shuffled source (compare Fig 5B) when also the target is shuffled. The red line indicates the median of the distribution of $\delta'_{TE}$. The black line indicates the original distance between the median of the surrogate data distribution with a shuffled source and the $mTE$ value computed on the original data. If this latter value is found in the upper rejection interval of the distribution of $\delta'_{TE}$, there is significant direct information transfer from the source to the target frequency band under investigation. (Left Panel) No information transfer remains when the source sending scale and the target receiving scale are simultaneously shuffled and no drop of $mTE$ can be seen (the distribution $\delta'_{TE}$ approaches 0); the original drop in $mTE$ is significantly larger. (Right panel) Information transfer remains when an unrelated target frequency band is shuffled. $\delta^j_{TE_{S_t}}$ (black bar), median of the $\delta'_{TE}$ distribution (red dotted bar).

**Table 1. Results of TE analysis.**

| simulation system | interaction<br>*source → target* | TE max lag<br>(ms) | *p*-values |
|---|---|---|---|
| *Example 1* | uncoupled | n/a | ns |
| *Example 2* | $S_0 \rightarrow T_1$ | 16 | <0.01** |
| *Example 3* | $S_0 \rightarrow T_1$ | 16.6 | <0.01** |
| *Example 4* | $S_{y0} \rightarrow T_{y1}$ | 8.3 | <0.01** |
| | $S_{y1} \rightarrow T_{y0}$ | | <0.01** |
| *Example 5* | $S_0 \rightarrow T_1$ | 4.1 | <0.01** |
| *Example 6* | $S_1 \rightarrow T_0$ | 4 | <0.01** |
| *Example 7* | $S_0 \rightarrow T_1$ | 16 | <0.01** |
| *Example 8* | $S_0 \rightarrow T_1$ | 8 | <0.01** |
| *Example 9* | $S_0 \rightarrow T_1$ | 4 | <0.01** |

$^*p < 0.05$;

$^{**}p < 0.01$;

$^{***}p < 0.001$;

**Table 2. Results of spectral TE analysis.**

| simulation system | simulated scale source → target | scale of maximum drop at source | scale of maximum drop at target | *p*-values |
|---|---|---|---|---|
| *Example 1* | n/a | n/a | n/a | n/s |
| *Example 2* | 4 →1 | 4 | 1 | <0.01** |
| *Example 2 SOSO* | 4 →1 | 4 | 1 | <0.05* |
| | 4 →2 | | | ns |
| *Example 3* | 1 →1 | 1 | 1 | <0.01** |
| *Example 4* | 4 →4 | 4 | 4 | <0.01* |
| | 1 →1 | 1 | 1 | <0.01* |
| *Example 4 SOSO* | 4 →4 | 4 | 4 | <0.05* |
| | 4 →1 | 4 | 1 | <0.05* |
| | 1 →1 | 1 | 1 | <0.05* |
| | 1 →4 | 1 | 4 | <0.05* |
| *Example 5* | 5 →1 | 5 | 1 | <0.01** |
| *Example 6* | 5 | 5 | | <0.01** |
| *Example 7* | 3, 4, 5 →1 | 3,4,5 | 1 | <0.01** |
| *Example 8* | 5 →1, 2, 3 | 5 | 1,2,3 | <0.01** |
| *Example 9* | 4 →2 | 4 | 2 | <0.01** |
| *Example 9 SOSO* | 4 →2 | 4 | 2 | ns |

*$p < 0.05$;

**$p < 0.01$;

***$p < 0.001$;

able to recover the information transfer in terms of identifying the correct frequencies via the scale of the maximal drop of the surrogate-based distribution and with statistical significance at those frequencies.

## Evaluation of the SOSO algorithm (II) on example 2

Here, we evaluated the SOSO algorithm on the CFIT from above (section *Example 2: Information transfer from one source to one target frequency in a bivariate system*). In the first SOSO analysis we set the source scale to be tested to $j = 4$ and the target scale to $j = 1$—as these were revealed by the spectral mTE analysis. As a control analysis, intended only for demonstration purposes here, the target scale was set to $j = 3$, i.e. a scale not identified as receiving information.

In the first SOSO analysis, the distance $\delta^4_{TE_{S_0}}$ was significantly bigger than the median of the distribution of the $\delta'_{TE}$ (see Fig 5C and Table 2), when we simultaneously destroyed source and target specific scale, indicating a direct information flow between the source and the target. In contrast, and as expected, no significant difference was found when we set the target scale to $j = 3$, since the simulated information transfer between source scale $j = 4$ and target $j = 1$ was not removed by shuffling at the wrong scale (i.e. $j = 3$). In section *Relation of the partial-information framework and the SOSO-algorithm* we further outline the importance of the SOSO algorithm in terms of the PID framework.

## Example 3: Comparison with Granger causality

In this example, we compared the spectral mTE algorithm with the spectral Granger causality. First, we simulated two AR(2) processes that exhibit autonomous oscillations at $f_1 = 45$ Hz and

are linearly coupled with a delay of 2 samples. We simulated 10 s at the sampling rate of 120 Hz and 100 trials (120000 samples). The AR(2) processes were generated as follows:

$$S_0(t) = 2p \; cos(2\pi f_1) S_0(t-1) - p^2 S_0(t-2) + \sigma_1 \eta_1(t) + w_1 \tag{14}$$

$$T_1(t) = 2p \; cos(2\pi f_2) T_1(t-1) - p^2 T_1(t-2) + c_1 S_0(t-2) + \sigma_2 \eta_2(t) + w_2 \tag{15}$$

where $c_1 = 0.7$ and determines the coupling strength, $p = 0.98$, $\eta_{1,2}$ are the innovation terms and $\sigma_1 = 0.3$, $\sigma_2 = 0.45$ control the strength of the innovation terms contribution. Additional white noise ($w$) was added as observation noise to the time-series. Since the model order was known, in this simulation, we used a parametric spectral Granger with a model order of two. Second, we compared the application of the nonparametric spectral Granger [34] with Example 2, where a CFIT occurred. We computed the cross-spectral density matrix using the fast fourier transform (FFTT) in combination with multitapers (2 Hz smoothing). To assess the significance level, we generated 500 permutations, by shuffling the trial order for the source, and obtaining the 95%-quantile of the maximal values over the frequencies (maximum statistics for multiple comparison correction). The null-hypothesis is that of no differences between the Granger frequency spectra of the coupled and the noncoupled system.

As before, we first recovered the simulated system connectivity with *mTE*, which yielded a significant information transfer from $S_0$ to $T_1$ at 1 sample lag (Table 1). Next, we applied the spectral *mTE* to obtain a frequency resolution of the system. Fig 6B left, shows a significant drop at the source site at scale 1 (frequency band $30 - 60$ Hz), containing the 45 Hz peak. The source frequency, linearly entering the target can be seen at the target site, with a significant drop at the same scale 1 (Fig 6B right). Then, we compared this result with the spectral Granger causality analysis. As shown in Fig 6C left, a significant interaction could be found from $S_0$ to $T_1$, with a Granger frequency peak at 45 Hz (blue line significantly above the 95% significance level, black dashed line). Next, we applied a nonparametric spectral Granger to

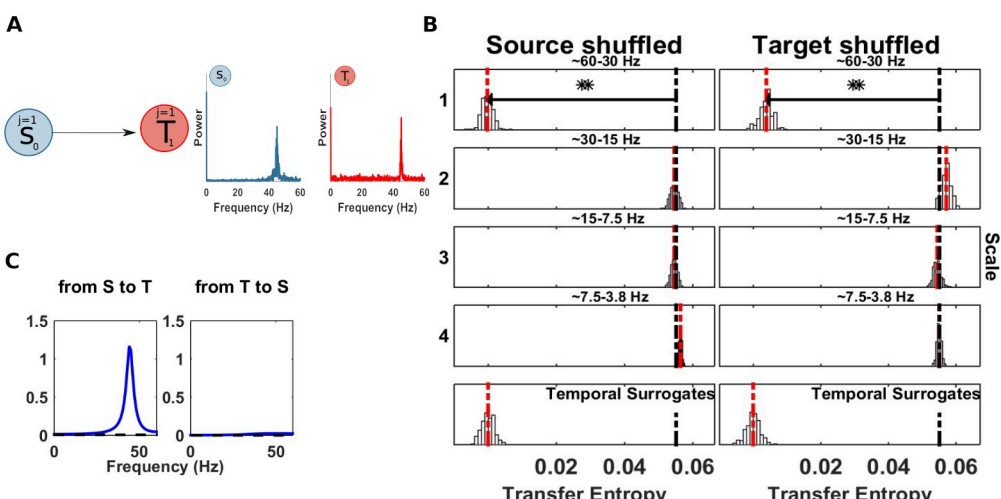

**Fig 6. Comparison of spectrally resolved transfer entropy to spectral Granger causality for within-band transfer (example 3).** (A) Left, a source $S_0$, is unidirectionally coupled, at scale j = 1 (frequency band $30 - 60$ Hz), with a target $T_1$. Right, power spectra of $S_0$ and $T_1$. (B) Spectrally resolved TE. Information transfer, correctly, drops when wavelet coefficients are selectively shuffled at scale 1 (frequency band 30-60 Hz) at the source site (left panel). At the target site the drop of wavelet coefficients at scale 1 exhibits the frequency entering the target linearly. (right panel). (C) Parametric spectral Granger analysis. First panel, a significant source was identified with peak at 45 Hz, (Granger causality estimates blue line). The 95% significance level obtained by permutation is indicated by a black dashed line).

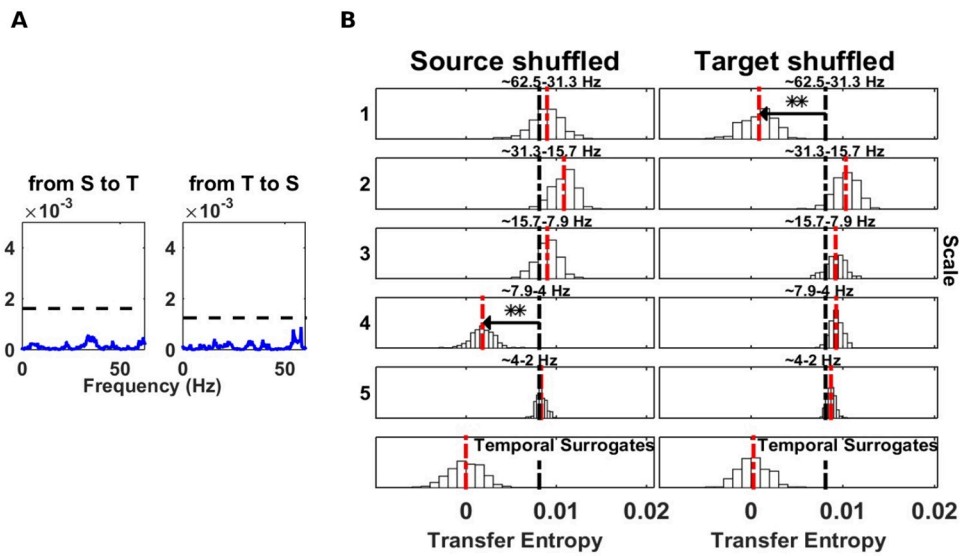

**Fig 7. Nonparametric spectral Granger causality in a system with cross-frequency information transfer (CFIT, example 2).** (A) Nonparametric spectral Granger causality. No significant source could be found in either directions. Granger causality estimates (blue line) and the 95% significance level obtained by permutation (black dashed line). (B) Spectrally resolved information transfer of the same system, in contrast, reveals the CFIT.

Example 2. As expected for a scenario with only CFIT, in this scenario the Granger analysis did not exhibit any significant result (Fig 7A).

## Example 4: Bidirectional system

In this example we simulated a bidirectional system of two AR(2) processes with different oscillatory profiles.

$$y_0(t) = 2p\ cos(2\pi f_1)y_0(t-1) - p^2 y_0(t-2) + c_1 y_1(t-1) + \sigma_1 \eta_1(t) + w_1 \qquad (16)$$

$$y_1(t) = 2p\ cos(2\pi f_2)y_1(t-1) - p^2 y_1(t-2) + c_2 y_0(t-1) + \sigma_2 \eta_2(t) + w_2 \qquad (17)$$

where: $f_1 = 5$ Hz, $f_2 = 45$ Hz, $p = 0.98$, $\eta_{1,2}$ are the innovation terms and $\sigma_1 = 0.6$, $\sigma_2 = 0.7$ control the strength of the innovation terms contribution and $c_1 = 0.3$, $c_2 = 0.2$ are the couplings parameters. Additional white noise ($w$) was added to the time-series. The simulation consisted of 20 seconds and 50 trials at the sampling rate of 120 Hz (120000 samples). First, we applied the multivariate $mTE$ to recover the system, this analysis showed a significant TE from $S_{y_0}$ to $T_{y_1}$ and, vice-versa $S_{y_1}$ to $T_{y_0}$ (see Table 1); here, $S_{y_0}$ indicates that process $Y_0$ is considered as the source, and $T_{y_1}$ means that process $Y_1$ is considered to be the target, and vice versa. Next, we applied the spectral $mTE$ to identify the relevant frequency bands.

The results in this system are interesting in two aspects. First, the expected within-band information transfer at the two different frequencies is recovered in both coupling direction, as expected, with the dynamic Eigen-frequency of the sending system correctly identified as the frequency of this transfer (Fig 8B and 8C, significant effects at the same scale for both columns and Table 2). Second, we see that the information of the sending process is received *in addition* at the dynamic Eigen-frequency of receiving process, i.e. at the frequency of the receiving process' dynamics when unperturbed by the coupling. We consider this result to be also correct as the innovation of the sending process is felt as an additional innovation term by

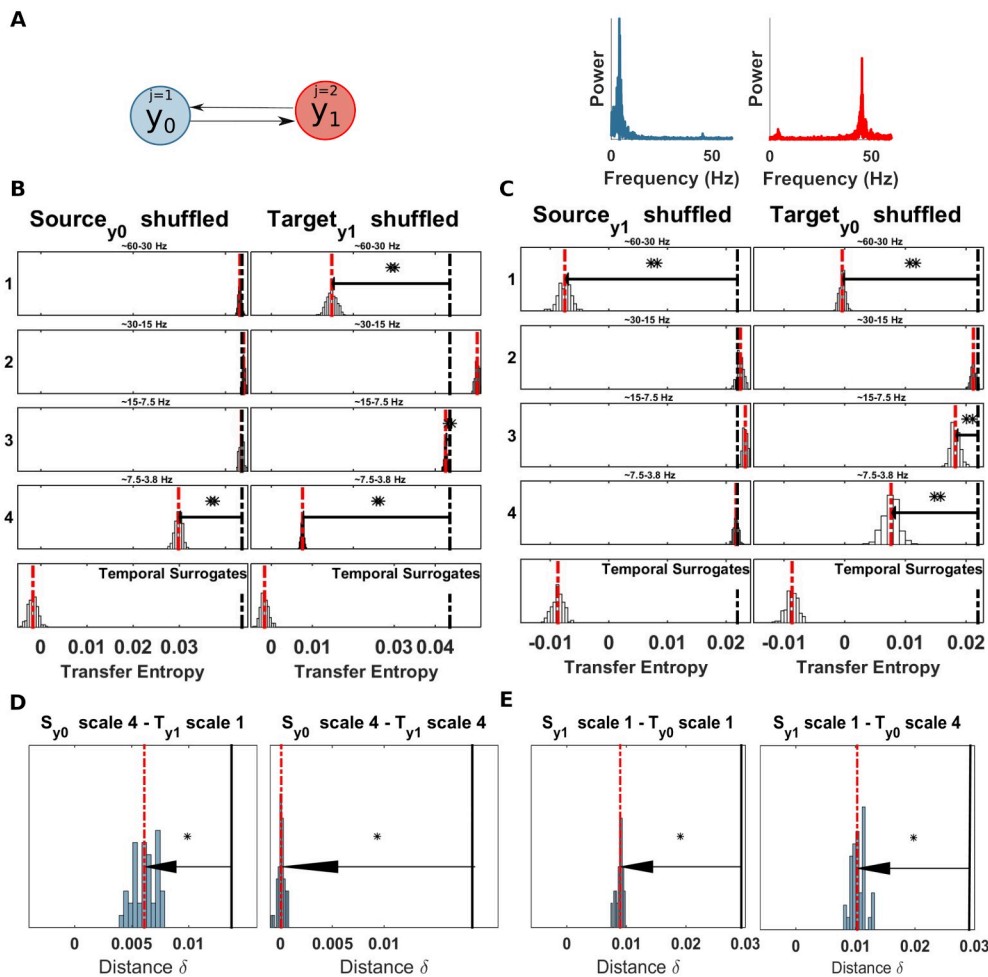

**Fig 8. Spectrally resolved transfer entropy for example 4.** (A) Left, a system with bidirectionally coupled nodes: $y_0$ and $y_1$. The process $y_0$ is linearly coupled with $y_1$ at scale j = 4 (frequency band 4-8 Hz) and the process $y_1$ is linearly coupled with $y_0$ at scale j = 1 (frequency band 30-60 Hz). Right, power spectral of $y_0$ and $y_1$. (B) Spectrally resolved Transfer Entropy for source $y_0$ and target $y_1$. See Fig 4 for display conventions. (Left panel) Information transfer, drops when wavelet coefficients are selectively shuffled at scale 4 (frequency band 4-8 Hz) on the source site. The corresponding reception of information at the target is shown on the right panel, where a drop for shuffled wavelet coefficients is also observed at scale 4 (frequency band 30-60 Hz). A significant drops is also observed at scale 1, in relation to the autonomous oscillations of the target. (C) Spectrally resolved Transfer Entropy for source $y_1$ and target $y_0$. See Fig 4 for display conventions. (Left panel) Information transfer, drops when wavelet coefficients are selectively shuffled at scale 1 (frequency band 30-60 Hz) on the source site. The corresponding reception of information at the target is shown on the right panel, where a drop for shuffled wavelet coefficients is also observed at scale 1 (frequency band 30-60 Hz). A significant drops is also observed at scale 4, in relation to the autonomous oscillations of the target. (D) SOSO analysis for source $y_0$ and target $y_1$ (E) SOSO analysis for source $y_1$ and target $y_0$. Both, within- and cross-frequency information transfer is detected. (For plotting conventions see Fig 5).

the receiving process in a 1-to-1 manner due to the additive linear coupling. Thus, the information generated in the innovations of the sender (e.g. system $y_0$) is incorporated into the dynamics of the receiver (e.g. system $s_1$). This is also supported by the SOSO analysis that does show a significant effect for the out-of-band analysis (Fig 8D and 8E). This is because the innovations produce information on a white spectrum, thus they do indeed have some information at the chosen sending frequency that is then incorporated at the receiving system's Eigen-frequency. This "cross-frequency" effect will be missed by classic within band analyses. We also stress that this second effect is not a failure of our approach in the bivariate setting.

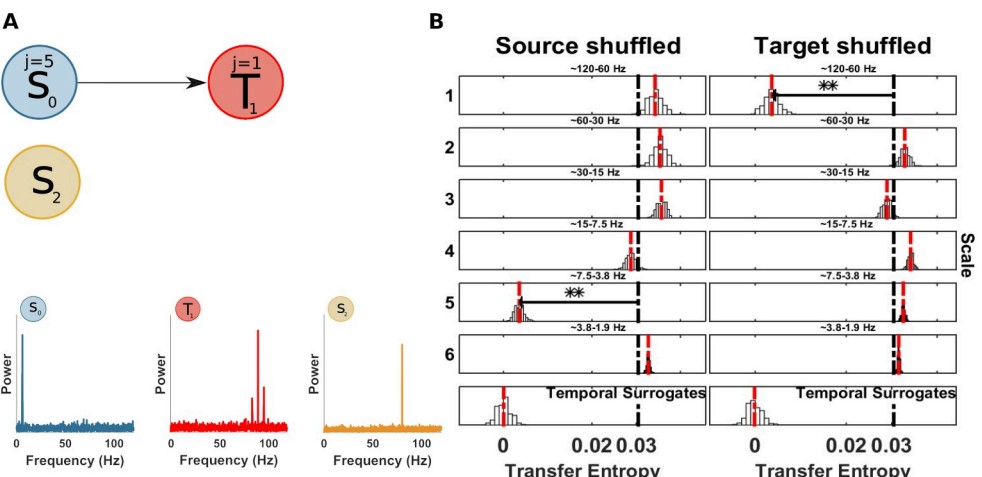

**Fig 9. Spectrally resolved transfer entropy for example 5.** (A) Top, a source $S_0$, but not $S_2$, is unidirectionally coupled, at scale j = 5 (frequency band 4-8 Hz), with a target $T_1$ at scale j = 1 (frequency band 60-120 Hz). Bottom, power spectral of $S_0$, $S_2$ and $T_1$. (B) Spectrally resolved Transfer Entropy. See Fig 4 for display conventions. (Left panel) Information transfer, correctly, drops when wavelet coefficients are selectively shuffled at scale 5 (frequency band 4-8 Hz) on the source site. The corresponding reception of information at the target is shown on the right panel, where a drop for shuffled wavelet coefficients is observed at scale 1 (frequency band 60-120 Hz), which contained the simulated target frequency.

## Example 5: Cross-frequency information transfer (CFIT) with nonlinear coupling in a multivariate three-node system

Next, we generated a multivariate network of three nodes. The network simulation was generated as follows: first, a source $S_0$ was coupled to a target with a CFIT. We adapted an example of [35] to simulate a more complex scenario with a non-sinusoidal driver, since in nonlinear systems such as the brain, perfect sinusoidal are often an exception [36]. The CFIT coupling was between $f_1$ = 6 Hz and $f_2$ = 80 Hz. To modulate the amplitude of the target time series we employed a sigmoid on the source and a delay of 2 samples. Second, to create a multivariate network, a 'distractor' node $S_2$ was added with an oscillation of $f_3$ = 90 Hz and it was modulated independently of the $f_1$ = 6 Hz of $S_0$ (Fig 9A). The simulation consisted of 10 seconds and 50 trials with a sampling rate of 240 Hz (120000 samples). The driver $S_0(t)$ was generated applying a bandpass filter to a Gaussian white noise at center frequency $f_1$ with a bandwidth of 0.4 Hz. The $S_2(t)$ and $T_1(t)$ were generated as follows:

$$S_2(t) = A * sin(2\pi f_3 t + \theta) + w_1 \tag{18}$$

$$T_1(t) = A * sin(2\pi f_2 t + \theta) * g(S_0(t-2)) + w_2 \tag{19}$$

where $g(x)$ is the sigmoid function:

$$g(x) = \frac{1}{1 + exp(-\lambda x(t))} \tag{20}$$

with $\lambda$ = 3. Gaussian white noise with a standard deviation of 1.2, 0.8, 0.6 was added to the signal $S_0$, $S_2$ and to the target $T_1$, respectively.

The TE analysis recovered the multivariate network with the associated delay (Table 1), identifying significant TE only between $S_0$ and $T_1$. The spectral mTE revealed the CFIT

between $S_0$ and $T_1$, with the maximal distance from the $mTE_{tot}$ at scale 5 for $S_0$ and scale 1 for $T_1$ (Fig 9B).

### Example 6: Delay-coupled Rossler systems (nonlinear)

We evaluated the spectral $mTE$ with a non-linear system able to generate self-sustained non-periodic oscillations. To this end we generated a coupled Rössler oscillator similar to [37]. The model was simulated with the following equations:

$$\frac{dx_1}{dt} = -w_1 y_1 - z_1 + \epsilon x_2(t - \tau) \qquad \frac{dx_2}{dt} = -w_2 y_2 - z_2$$

$$\frac{dy_1}{dt} = w_1 x_1 + 0.15 y_1 \qquad \frac{dy_2}{dt} = w_2 x_2 + 0.15 y_2 \qquad (21)$$

$$\frac{dz_1}{dt} = 0.2 + z_1(x_1 - 10) \qquad \frac{dz_2}{dt} = 0.2 + z_2(x_2 - 10)$$

where $w_1$ and $w_2$ are the natural frequencies of the oscillator which were set to 0.8 and 0.9, $\epsilon = 0.07$ is the coupling strength and $\tau$ is the time delay, which was set to 2 time steps. As can be seen in Fig 10A, the two systems oscillated around 8 Hz, but were not identical. The analysis was performed on the assumption that only variables $x_1(t)$ and $x_2(t)$ could be observed, with $S_1 = x_2(t)$ and $T_0 = x_1(t)$, in this simulation. The sampling rate was 500 HZ, 4 seconds and 50 trials were generated (200000 samples).

The TE analysis correctly identified the driver $S_1$ with a delay of 2 samples (Table 1). The spectral mTE showed a significant drop at scale 5 (Fig 10B). No significant drop was observed at the target site $T_0$.

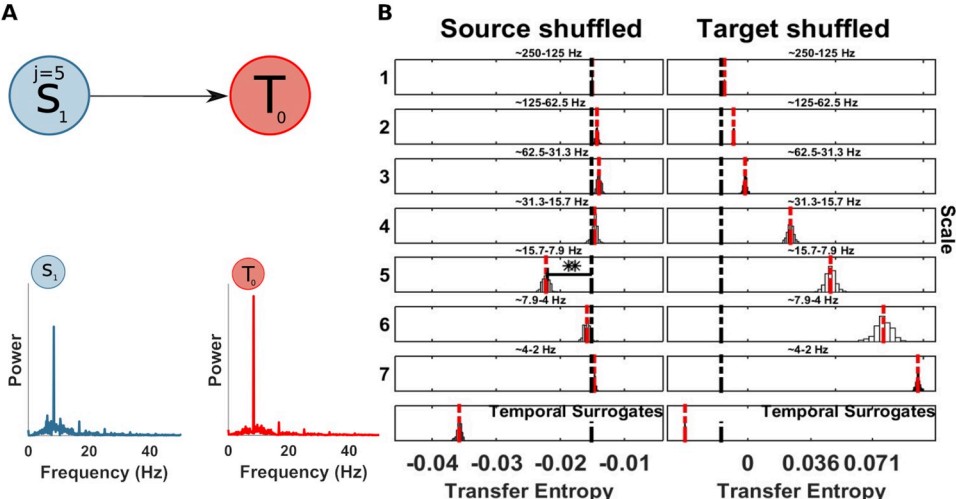

**Fig 10. Spectrally resolved transfer entropy for example 6.** (A) Top, a source $S_1$ is unidirectionally coupled, at scale $j = 5$ (frequency band 8-16 Hz), with a target $T_0$. Bottom, power spectra of $S_1$ and $T_0$. (B) Spectrally resolved Transfer Entropy. See Fig 4 for display conventions. (Left panel) Information transfer, correctly, drops when wavelet coefficients are selectively shuffled at scale 5 (frequency band 8-16 Hz) on the source site. (Right panel) No significant drop is present at the target site.

### Example 7: Information transfer from multiple source frequencies to one target frequency

To test the ability of the spectral $mTE$ to recover multiple source frequencies sending information to a target frequency, we simulated a bivariate example similar to *Example 1* but with multiple source scales showing a phase-amplitude relation with a single target scale.

$$S_0(t) = A(cos(2\pi f_1 t + \theta) + cos(2\pi f_2 t + \theta) + cos(2\pi f_3 t + \theta)) + w_1 \tag{22}$$

$$T_1(t) = A * cos(2\pi f_4 t + \theta) * S_0(t - 2) + w_2 \tag{23}$$

with $S_0$ constructed as a sum of sinusoids with different phases $\theta$ and Gaussian noise processes for $w_1$ and $w_2$, as before. Then, the source $S_0$ modulates the amplitude of the target $T_1$ at scale $j = 1$ with a sample delay of 2 (Fig 11A). The simulation consisted of 5 seconds with 50 trials at 125 Hz (31250 samples).

The TE analysis showed the source $S_0$ as driver of the target $T_1$ (Table 1), as simulated with a sample delay of 2. Then, we applied the Spectral TE to identify the three scales of the source $S_0$ sending information to the target scale $j = 1$. Fig 11B, showed three significant scales: 3 (frequency band 8-16 Hz), 4 (frequency band 4-8 Hz), 5 (frequency band 2-4 Hz), at source site and scale 1 (frequency band 31-63 Hz) at target site. This application demonstrates the ability of the spectral TE to recover multiple sources sending information to the target site. We also noted that the three scales at the source have a slightly different drop (or maximal distance from original $mTE_{tot}$) which can be related to how noise affects the wavelet frequency decomposition at different scales.

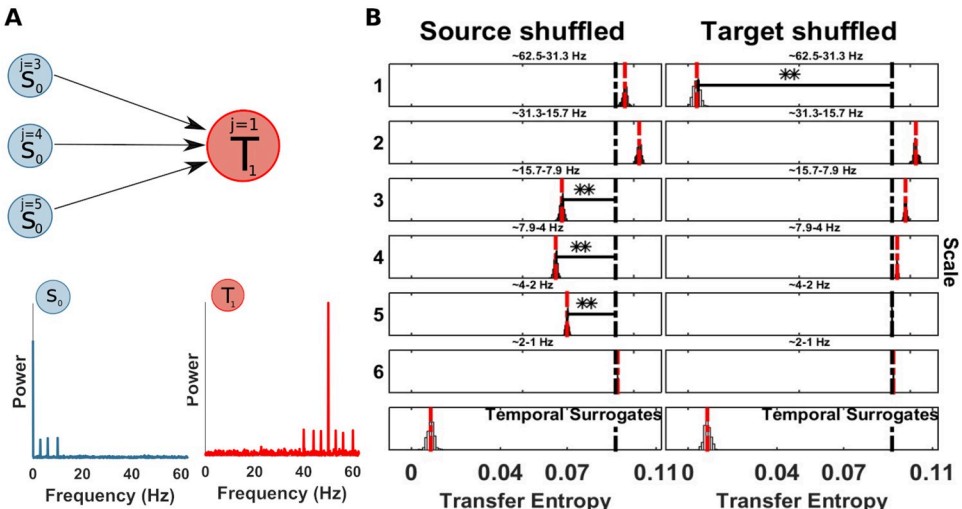

**Fig 11. Spectrally resolved transfer entropy for example 7.** (A) Top, a source $S_0$ is unidirectionally coupled, at multiple scales: j = 3 (frequency band 8-16 Hz), j = 4 (frequency band 4-8 Hz) and j = 5 (frequency band 2-4 Hz), with a target $T_1$ at scale j = 1 (frequency band 31-63 Hz). Bottom, power spectral of $S_0$ and $T_1$. (B) Spectrally resolved Transfer Entropy. See Fig 4 for display conventions. (Left panel) Information transfer, correctly, drops when wavelet coefficients are selectively shuffled at scale 3 (frequency band 8-16 Hz), 4 (frequency band 4-8 Hz), 5 (frequency band 2-4 Hz) on the source site. The corresponding reception of information at the target is shown on the right panel, where a drop for shuffled wavelet coefficients is observed at scale 1 (frequency band 31-63 Hz).

### Example 8: Information transfer from one source frequency to multiple target frequencies

To test the ability of the spectral $mTE$ to recover one source frequency sending information to multiple target frequencies, we simulated a bivariate example with a single source frequency showing a phase-amplitude relation with multiple target frequencies.

$$S_0(t) = A * cos(2\pi f_1 t + \theta) + w_1 \tag{24}$$

$$y(t) = A * (cos(2\pi f_2 t + \theta) + cos(2\pi f_3 t + \theta) + cos(2\pi f_4 t + \theta)) \tag{25}$$

$$T_1(t) = y * S_0(t - 1) + w_2 \tag{26}$$

where $y$ is a sum of sinusoids at different high frequencies, with $f_2 = 100$ Hz, $f_3 = 58$ Hz and $f_4 = 30$ Hz, which are modulated by the source $S_0$ with $f_1 = 5$ Hz, with a sample delay of 1 (Fig 12A), and different noise levels, $w_1$ with a standard deviation of 0.4 and $w_2$ with a standard deviation of 1. The simulation consisted of 5 seconds with 50 trials at 250 Hz (62500 samples).

The TE analysis correctly identified the source $S_0$ as the driver of $T_1$ with the sample delay of 1 (Table 1). The application of the spectral TE showed two significant scales at the source site: 5 (frequency band 4-8 Hz) and 6 (frequency band 2-4 Hz). In this application, the simulated source scale ($j = 5$) is recovered by the spectral analysis and it is the scale with the largest drop (see Fig 12B). However, scale 6 is also significant. At the target site three scales have a significant drop: 1 (frequency band 63-125 Hz), 2 (frequency band 31-63 Hz) and 3 (frequency band 16-31 Hz), as simulated. This application showed the ability of the spectral TE algorithm to identify multiple receiving targets from one source. Although, the scale with the largest drop, reliably reflects the ground truth, nearby scales have to be interpreted with caution,

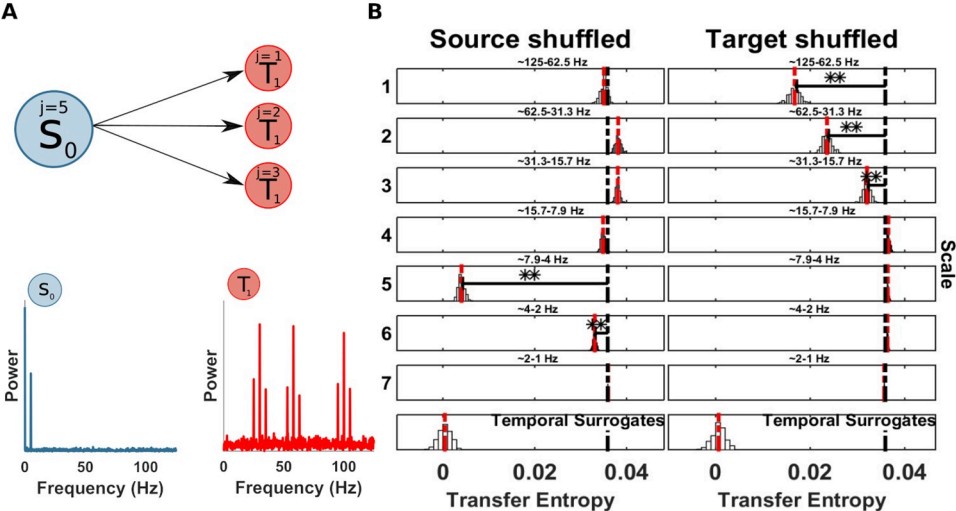

**Fig 12. Spectrally resolved transfer entropy for example 8.** (A) Top, a source $S_0$ is unidirectionally coupled, at scales $j = 5$ (frequency band 4-8 Hz), with a target $T_1$ at multiple scales: $j = 1$ (frequency band 63-125 Hz), $j = 2$ (frequency band 31-63 Hz) and $j = 3$ (frequency band 16-31 Hz). Bottom, power spectrum of $S_0$ and $T_1$. (B) Spectrally resolved Transfer Entropy. See Fig 4 for display conventions. (Left panel) Information transfer, drops when wavelet coefficients are selectively shuffled at scale 5 (frequency band 4-8 Hz) and 6 (frequency band 2-4 Hz) on the source site. The corresponding reception of information at the target is shown on the right panel, where a drop for shuffled wavelet coefficients is observed at scale 1 (frequency band 63-125 Hz), scale 2 (frequency band 31-63 Hz) and scale 3 (frequency band 16-31 Hz).

which might have a substantial smaller drop, but still significant, if the the spectral decomposition is not perfectly confined within a single band. To obtain a better frequency concentration, we repeated the analysis employing the MODWT with LA(16). Using a longer wavelet filter should decrease the spectral leakage at nearby scales although increasing the number of boundary-coefficients. Indeed, this analysis revealed a correct identification of the only simulated scale 5 (frequency band 4-8 Hz) at the source site, see *Supplementary Material* (S1 Fig).

### Example 9: Multiple information flows at multiple frequencies from source to target

In this final simulation, we tested the ability of the SOSO algorithm to rule out direct information flow from a source to a target when one source frequency sends information redundantly into all target frequencies, while one target frequency receives (other) information redundantly from all source frequencies (see Fig 13A, and also Fig 2). The simulation consisted of 10 seconds at the sampling rate of 250 Hz and 50 trials (125000 samples) and it was carried out according to:

$$S_0(t) = x_1 + x_2 + w_1 \tag{27}$$

$$T_1(t) = y_1 + y_2 + w_2 \tag{28}$$

where $x_1$ and $y_1$ were simulated with:

$$x_1(t) = A^* * cos(2\pi f_1 t + \theta) \tag{29}$$

$$z(t) = \sum_{i=2}^{l} A * cos(2\pi f_i t + \theta) \tag{30}$$

$$y_1(t) = z * x_1(t - 1) \tag{31}$$

where $l = 6$ and with $f_1 = 9$ Hz ($j = 4$), $A^*$ set to 2, $f_2 = 80$ Hz ($j = 1$), $f_3 = 40$ Hz ($j = 2$), $f_4 = 18$ Hz ($j = 3$), $f_5 = 9$ Hz ($j = 4$) and $f_6 = 5$ Hz ($j = 5$). The signals $x_2$ and $y_2$ were simulated with:

$$x_2(t) = \sum_{i=2}^{l} A * cos(2\pi g_i t + \theta) \tag{32}$$

$$y_2(t) = A * cos(2\pi g_1 t + \theta) * x_2(t - 1) \tag{33}$$

where $l = 6$ and with $g_1 = 40$ Hz ($j = 4$), $g_2 = 80$ Hz ($j = 1$), $g_3 = 40$ Hz ($j = 2$), $g_4 = 18$ Hz ($j = 3$), $g_5 = 9$ Hz ($j = 4$) and $g_6 = 5$ Hz ($j = 5$). The parameter $\theta$ is a uniform random variable between 0 and $2\pi$, $w_1, w_2$ are samples of i.i.d Gaussian noise process with a standard deviation of 1, and all parameters $A$, except $A^*$, are set to 1.

The spectral TE analysis showed (Fig 13B), correctly, the largest drop at scale 4 (frequency band 8-16 Hz) at the source site (it was simulated with higher amplitude $A^* = 2$), because this is the source of non-redundant information; additionally scale 5 (frequency band 4-8 Hz) was significant, likely due to spectral leakage. No other scales could be detected at the source site. At the target site, the largest drop was at scale 2 (frequency band 31-63 Hz), being the scale coupled with source scale 4 but also receiving information from all others source scales, i.e. receiving multiple non-redundant information streams. Additionally, only scale 1 (frequency band 63-125 Hz) and 3 (frequency band 16-31 Hz) were significant at the target site. This example was designed to show that a direct non-redundant information transfer from source

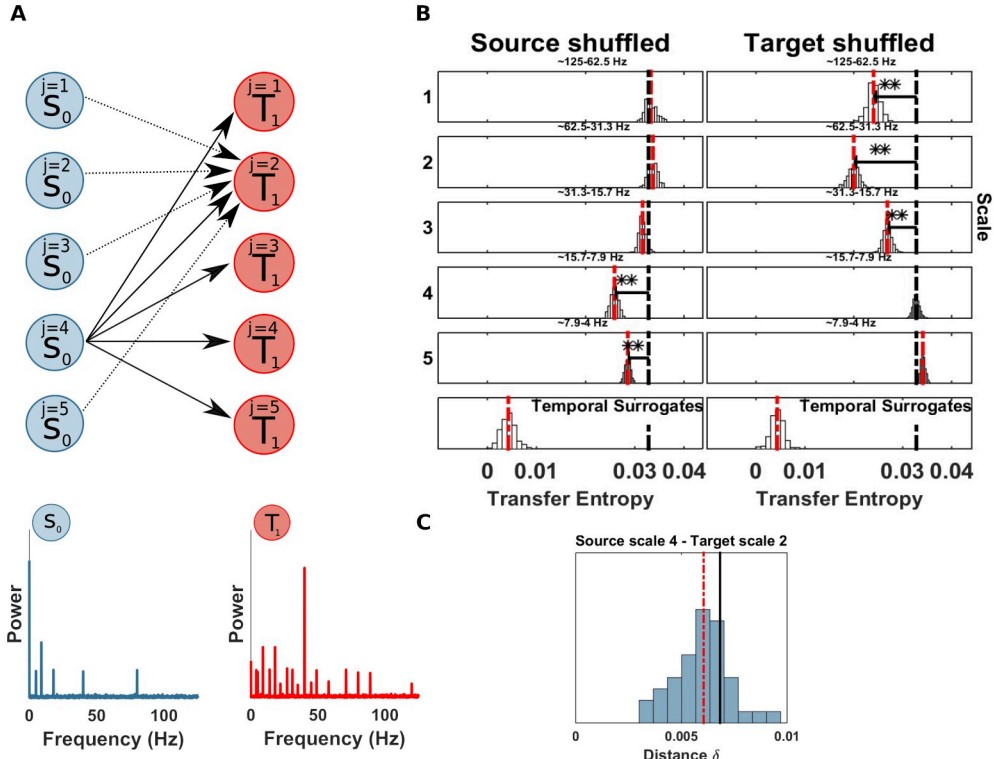

**Fig 13. Spectrally resolved transfer entropy for example 9.** (A) Top, a source $S_0$ is unidirectionally coupled, with target $T_1$. Multiple scales ($j$ = 1, 2, 3, 5) of $S_0$ are coupled with a single target scale 2 and at the same time a single source scale 4 of $S_0$ is coupled with multiple target scales ($j$ = 1, 2, 3, 4, 5). Bottom, power spectrum of $S_0$ and $T_1$. (B) Spectrally resolved Transfer Entropy. See Fig 4 for display conventions. (Left panel) Information transfer drops when wavelet coefficients are selectively shuffled at scale 4 (frequency band 8-16 Hz) and 5 (frequency band 4-8 Hz) on the source site. The corresponding reception of information at the target is shown on the right panel, where a drop for shuffled wavelet coefficients is observed at scale 1 (frequency band 63-125 Hz), scale 2 (frequency band 31-63 Hz) and scale 3 (frequency band 16-31 Hz). (C) SOSO application to redundant information flow. See Fig 5C, for display conventions. Information transfer remains when the source scale 4 and the target scale 2 are simultaneously shuffled, ruling out a direct information transfer between these two frequency bands.

scale 4 to target scale 2 can be ruled out by using the SOSO-Algorithm. Indeed, the SOSO algorithm confirmed that no direct, non-redudant information transfer took place (see Fig 13C). We direct the reader to section *Relation of the partial-information framework and the SOSO-algorithm*, for further discussion of the SOSO-algorithm in relation to the PID framework.

## Application to neural data

Finally, we tested the spectral TE method on two different neurophysiological datasets, first, a human MEG dataset with significant TE between seven sources published in [38], and, second, a local field potential (LFP) recording in the ferret in Prefrontal Cortex (PFC) and Primary Visual Area (V1) published in [39].

**Information transfer in MEG data from a Mooney face detection task.** The data analyzed here were published in [38]. In short, neural activity was recorded with MEG from n = 52 subjects at 1.2 kHz sampling rate on a 275 channel whole head magnetoencephalograph (Omega 2005, VSM MedTech). Subjects had to detect either faces (face condition) or houses (house condition) in a stream of black and white pictures of faces (Mooney faces), houses, and scrambled versions of these. From the MEG recordings task-relevant sources were identified

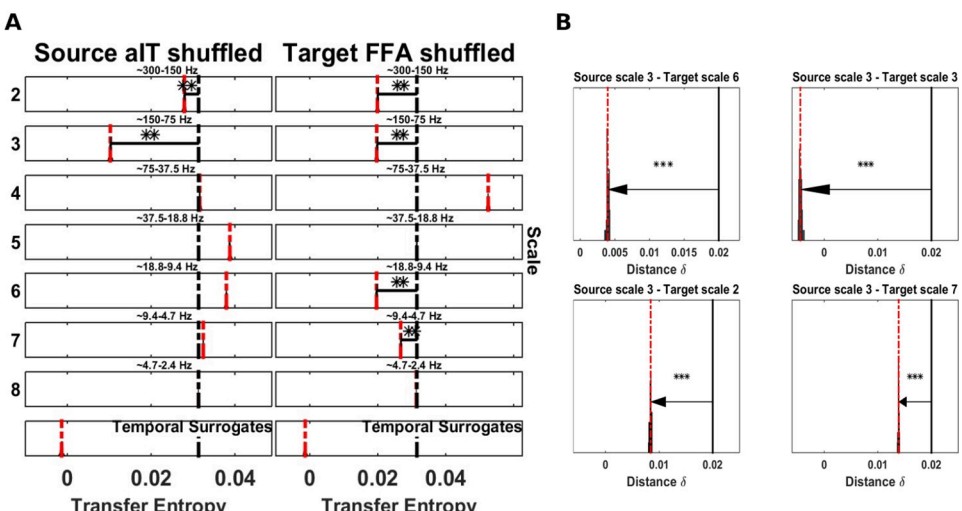

**Fig 14. Spectrally resolved information transfer between MEG sources when preparing to detect faces.** (A) Spectrally resolved information transfer between aIT as a source and FFA as a target in the condition where subjects are trying to detect target faces. aIT sends information mainly at 75-150Hz (left column), whereas FFA receives information at high frequencies (75-150Hz and above) as well as low frequencies (9-19Hz and 5-9Hz) (right column). See Fig 4 for display conventions. (B) Analyses of cross-frequency information transfer between specific source frequency in aIT and multiple target-frequencies in FFA (second panel on the right side). All four scales (2, 3, 6, 7) at the target side showed a significant direct information transfer from the source at scale 3. $\delta^{j}_{TE_{S_i}}$ (black bar), median of the $\delta'_{TE}$ distribution (red dotted bar).

by means of beamformer source reconstruction from pre-stimulus baseline data, reflecting the subject's expectations relevant for the original study, and a comparison of the local active information storage values between the two experimental conditions. After identifying 5 task-relevant sources, bivariate TE was computed on the baseline interval between all pairs of sources and compared between conditions. This procedure identified a significantly different TE between the two conditions (face, house conditions) from anterior inferotemporal cortex (aIT) to the fusiform face area (FFA). Here we follow up on these results by analyzing which source and target frequencies carried the TE found in the original study.

At the group level, for each scale in the face condition, we determined if the $m\tilde{T}E'$ was significantly smaller than the $mTE_{tot}$ (randomization test [40] on the dependent-samples t-statistic, alpha level was set at 0.05 and Bonferroni corrected for the fourteen scales tested).

In the face condition (Fig 14A), we found the aIT source to be sending information at frequencies around 110 Hz (scale 3: 75-150 Hz, with possible additional contributions above 150 Hz scale 2: 150-300 Hz), while the FFA target received significant information transfer at multiple high frequency bands: above 150Hz (scale 2: 150-300 Hz) and around 110 Hz (scale 3: 75-150 Hz) and multiple lower bands: around 14 Hz (scale 6: 9-19 Hz) and around 7 Hz (scale 7: 5-9 Hz). Results for the spectrally resolved mTE in the house condition were qualitatively similar (see S2 Fig).

Finally, we applied the SOSO algorithm to the face condition at scale 2 for the source aIT and to the four significant scales at the target FFA (scales: 2,3,6 and 7). The SOSO algorithm showed a significant direct information flow between source scale 2 (frequency band 75-150 Hz) and all four scales at the target site (scales: 2,3,6 and 7), confirming a high complexity of interaction between the source aIT and multiple frequencies at the FFA area (see Fig 14B).

These spectral mTE results provide important information about the spectral complexity of the interaction between aIT and FFA in that the information transfer took place between

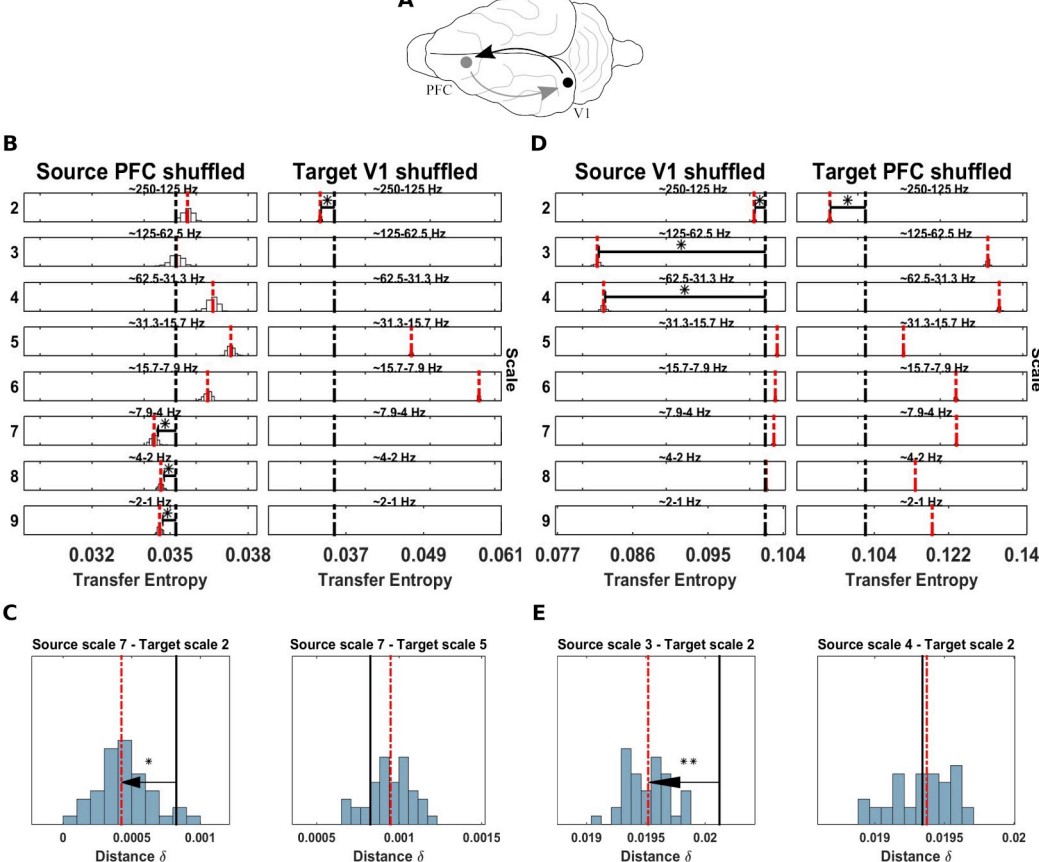

**Fig 15. Spectrally resolved information transfer for the ferret.** See Fig 4 for display conventions for B and D and Fig 5 for C and D. (A) Schematic location of recording sites on the ferret brain (from [39]). (B) Spectrally resolved information transfer from PFC to V1 in the ferret. (Left panel) Information transfer, drops at scale 7, 8, and 9 on the source site (PFC), when the wavelet coefficients are shuffled. (Right panel) A significant drop is observed at scale 2 at the target site (V1). $mTE_{tot}$ original (black dotted bar), $m\tilde{T}E^*$ (red dotted bar). Scale 1 is not shown since LFP were low passed at 300 Hz. (C) Analyses of cross-frequency information transfer between specific source- and target-frequencies in PFC and V1 of the ferret. (Left Panel) No information transfer is present when the source sending scale and the target receiving scale are simultaneously shuffled and no drop of $mTE$ can be seen (the distribution $\delta'_{TE}$ approaches 0). (Right panel) Information transfer is still present when unrelated target frequency band is shuffled. $\delta^j_{TE_{S_i}}$ (black bar), median of the $\delta'_{TE}$ distribution (red dotted bar). (D) Spectrally resolved information transfer from V1 to PFC. (Left panel) Information transfer, drops at scale 2, 3, and 4 on the source site (V1), when the wavelet coefficients are shuffled. (Right panel) A significant drop is observed at scale 2 at the target site (PFC). $mTE_{tot}$ original (black dotted bar), $m\tilde{T}E^*$ (red dotted bar). Scale 1 is not shown since LFP were low passed at 300 Hz. (E) SOSO application to source V1 and target PFC of ferret.(Left Panel) No information transfer is present when the source sending scale and the target receiving scale are simultaneously shuffled and no drop of $mTE$ can be seen (the distribution $\delta'_{TE}$ is centered on 0). This means that there is indeed a direct transfer of information from source scale 3 to target scale 2 (Right panel) Information transfer into target scale 2 is still present when the source scale 4 is shuffled, meaning information does not flow from source scale 4 to target scale 2. $\delta^j_{TE_{S_i}}$ (black bar), median of the $\delta'_{TE}$ distribution (red dotted bar).

different frequency bands. These results could not have been obtained with a spectral GC approach, as spectral GC only searches for within-band transfer of information.

**Occipito-frontal and fronto-occiptal information transfer in the ferret.** We applied the spectral TE method to data from a previous study on anaesthesia effects in the ferret [39]. In short local field potentials were recorded simultaneously in primary visual cortex (V1) and the prefrontal cortex (PFC) of two female ferrets (see Fig 15A, for a schematic depiction of

**Table 3. Results of TE analysis neural data.**

| experiment | interaction | significant subjects | *p*-values |
|---|---|---|---|
| *MEG face condition* | $aIT \rightarrow FFA$ | 29 | $<0.01^{**}$ |
| *LFP Ferret* | $PFC \rightarrow V1$ | n/a | $<0.05^{*}$ |
| | $V1 \rightarrow PFC$ | n/a | $<0.05^{*}$ |

$^{*}p < 0.05;$

$^{**}p < 0.01;$

$^{***}p < 0.001;$

recording sites) under different concentrations of the anesthetic isoflurane and under awake conditions. Since the application of spectrally resolved mTE here served only as a proof of principle we only analysed the data of ferret 1, i.e. the ferret that showed significant bi-directional TE in the awake condition. Moreover, we only analyzed data from the awake condition. For more information on the experimental procedures see [39]. First, using the mTE-algorithms from our new IDTxl toolbox we replicated the earlier findings of a significant bidirectional TE between PFC and V1 and vice versa (see Table 3).

Second, in the direction from PFC to V1 the spectral TE method revealed a significant effect in scales 7, 8, 9, with scale 7 (4-8 Hz) as minimum, when PFC was considered as source of V1. In contrast, at the target site (V1), the only scale that revealed a significant TE decrease when shuffled was scale 2 (125-250 Hz, high gamma band, or very high frequency oscillations, VHF), indicating a possible CFIT from PFC to V1, results are reported in Table 4. To test this, we applied the SOSO algorithm on scale 7 from the PFC source and scale 2 from the target V1. As a control analysis we additionally randomly picked another scale on the target side to verify that the interaction was restricted only to target scale 2.

Similarly to our simulations (see Fig 5C), the PFC source at scale 7 and the V1 target at scale 2 showed a significant decrease of the distribution of delta values and $\delta_{TE_{S_i}}$ was in the extreme 5% of the distribution ($p<0.05^{*}$). No significant result was found when we applied the SOSO algorithm to the randomly chosen control target-scale 5 (see Fig 15C left, and Table 4).

**Table 4. Results of spectral TE analysis.**

| experiment | scale of maximum drop at source | scale of maximum drop at target | *p*-values |
|---|---|---|---|
| *MEG faces condition* | 3 | 2,3,6,7 | $<0.01^{**}$ |
| *SOSO* | | | |
| $aIT \rightarrow FFA$ | 3 | 2,3,6,7 | $<0.001^{***}$ |
| *LFP Ferret* | | | |
| $PFC \rightarrow V1$ | 7 | 2 | $<0.05^{*}$ |
| $V1 \rightarrow PFC$ | 3 | 2 | $<0.05^{*}$ |
| *SOSO* | | | |
| $PFC \rightarrow V1$ | 7 | 2 | $<0.05^{*}$ |
| | 7 | 5 | ns |
| $V1 \rightarrow PFC$ | 3 | 2 | $<0.05^{*}$ |
| | 4 | 2 | ns |

$^{*}p < 0.05;$

$^{**}p < 0.01;$

$^{***}p < 0.001;$

Third, we considered V1 as source and PFC as target. The spectral TE algorithm revealed a significant TE decrease on scale 2, 3, 4 at source site, with scale 3 (high gamma) as minimum. On the target side (PFC) scale 2 was the only significant result (see Table 4).

We applied the SOSO algorithm on the source scale 3 and 4, and target scale 2. This analysis revealed a significant decrease of the distribution of delta values for source scale 3 and target scale 2 ($p<0.05^*$, see Fig 15E, left and Table 4), but interestingly not for source scale 4 (see Fig 15E, right and Table 4).

The application of the new spectral TE algorithm on data that showed significant bidirectional TE revealed, a low frequency top-down communication, PFC→V1, with a possible CFIT (Fig 15B), and high frequency bottom-up communication V1→*PFC* (Fig 15D), in agreement with [41].

These results demonstrate the value of separate analyses for source and target frequencies and the post-hoc tests to identify only direct TE from source to target.

## Discussion

We present an algorithm to measure the information transfer that is associated with specific spectral components in an information source or target. Our evaluation results demonstrate that spectral components in the source or target can be reliably identified, given that there are not many closely overlapping components contributing to the information transfer. If many, closely overlapping components are present, a conservative approach is to focus only on the component yielding the largest contribution. One of the advantages of the algorithm presented here is that the original signals are never filtered before the computation of information transfer. Rather, we defer the filtering operation to the creation of spectrally-specific surrogate data, where phase shifts, filtering artefacts or filter-inefficiencies only lead to overly conservative results in the worst case, but not to spurious false positives. Our algorithm can be extended to investigate whether the information transferred from a specific source-frequency indeed arrives at a specific target frequency, whenever there is an *a priori* reason to assume that this possibility exists in the system under investigation. If no such *a priori* consideration applies, the intricacies of the partial information decomposition warrant utmost caution when inferring a direct transfer of information from identified source to target frequencies (see the section on partial information decomposition, below).

In the remainder of this section we will detail the added value of spectrally-resolved TE analysis for neuroscience, discuss the inherent but often-overlooked complexities of measuring spectrally-resolved information transfer, discuss advantages and drawbacks of the algorithm presented, discuss its relation to previous approaches, and remark on the choice for the free parameters of the algorithm.

### Information transfer in rhythmic neural processes beyond within-band synchronization and linear interactions

Having the possibility to analyze how information is transferred in relation to various spectral components is of particular interest in neuroscience. This is because of the importance of information transfer for the distributed computation performed in neural systems and because of the prevalence of rhythmic processing in neural systems. This rhythmic processing is evident for example in the mammalian neocortex, where researchers have studied processes in various frequencies for decades (e.g. $\delta$, $\theta$, $\alpha$, $\beta$, low and high frequency $\gamma$ rhythms). Our novel analysis techniques offer the unique opportunity to unravel how information is exchanged between different rhythmic processes, but also, and importantly between arhythmic (wideband) and rhythmic processes (see *Example 7: Information transfer from multiple source*

*frequencies to one target frequency*). We thus expect that our methods will widen the current focus on synchronization and within-band interactions to reveal a fuller picture of neural processing.

In this respect, even our proof of concept analyses of MEG and LFP data have provided intriguing new insights:

1. In the MEG data we found two surprising results: (A) Information transfer at very high frequencies that was possibly linked to leaked multi-unit activity or to oscillatory components. We are not aware of other reports of functional connectivity or information transfer at those frequencies, possibly owing to the fact that coupling at these frequency bands may be nonlinear and may not be carried by relatively stable oscillations. Nevertheless our analyses demonstrate that the information in these bands is well captured by the MEG. (B) The information sent via high frequencies from aIT cortex to FFA is also received at the low beta band, i.e. there is information flowing from high frequencies to lower frequencies. This result differs from the usual assumption that lower frequencies modulate higher ones (e.g. via phase-amplitude coupling [31]), but corroborates earlier findings in nonhuman primates that showed the same effect [3].

2. In the ferret LFP-recordings, we observed information being sent from frontal cortices at low frequencies from $\delta$, $\theta$ and $\alpha$-bands, and being received in V1 at high $\gamma$-frequencies—in line with previous reports. Yet, of the information in the low source frequencies, only information in the alpha band seems to be directly received by the high $\gamma$-band in the target—with the information sent by the $\delta$ and $\theta$-bands being either redundant with the information in the $\alpha$-band or being received across all frequencies in the target as indicated by the SOSO-analysis (Fig 15C).

In sum, these two exemplary applications to neural data demonstrate the enormous level of detail provided by the proposed algorithms for the analyses of neural communication. These examples also point to the possibility that there are many neural communication processes or mechanisms that have been overlooked so far due to the lack of proper analysis methods.

## Frequency resolved TE as a partial information decomposition problem

To discuss frequency-resolved analysis of information transfer as a partial information decomposition (PID) problem, we first introduce the concept of PID by a simple example. Imagine two source variables $S_1$, $S_2$ that together provide some information about a target variable $T$, i.e. the joint mutual information $I(T: S_1, S_2)$ is non-zero. One may ask, then, how much of that joint mutual information about $T$ can only be obtained from $S_1$, but not from $S_2$ and vice versa, how much information can be redundantly obtained from either variable, and how much information can be only obtained from $S_1$, $S_2$ considered jointly. These three 'types' of information are called unique, shared and synergistic mutual information (see [42, 43] for reviews, and [8, 9, 44, 45] for further details).

In the same way that a joint mutual information can be decomposed we can also decompose a conditional mutual information, e.g. $I(T: S_1, S_2|Z)$. Since transfer entropy is just a conditional mutual information, and since our spectral components would take the role of the source variables $S_1, \ldots, S_n$ we see that asking for the contribution of each spectral component to the overall transfer entropy amounts to solving the partial information decomposition problem. Unfortunately, the full complexity of such partial-information decompositions has only been realized very recently, and their mathematical formulation is still a matter of active research (see for example the recent special issue on this topic [46]). Yet, independent of the specific mathematical formulation, the PID framework is a useful tool to think about the

inherent complexity of spectrally-resolved information transfer in order to better interpret results. We would like to stress, however, that our proposed algorithms in no way depend on a solution to the PID problem, as all quantities involved are just classic conditional mutual information terms.

At present, existing PID measures only allow a decomposition of either the source or the receiver processes into PID components. This unsolved problem is also the fundamental reason why we have mostly confined ourselves to considering source and receiver frequencies separately (apart from trying to identify the special case of one source frequency interacting with one target frequency). While the field of PID is still under rapid development in terms of proper information theoretic measures, the underlying structure of the problem is clear, and can be harnessed to understand the spectral analysis of information transfer. In particular, using the PID formalism we can clearly map out which specific components and combinations of components of a PID will be detected or missed by our analysis method, irrespective of any particular definition or measure of PID components:

1. Frequencies on the source or target side that contribute unique information to the transfer entropy from source to target will be detected, both on the source and the target side.

2. Frequencies on the source or target side that carry information redundantly (and with approximately equal amount and signal to noise ratio) will be *missed*. This is because destroying one of these frequencies in the surrogate data will not remove the information, as it is redundantly carried via the other frequency. Hence, the mTE on the permuted data will most likely not drop significantly.

3. Frequencies on the source or target side that synergistically contribute to the TE will all be detected, as destroying each of them will reduce the amount of information transfer in the permuted data. There will be no indication as to whether the information transfer was a synergistic or unique contribution of those frequencies, unless there is only one frequency that leads to a reduction of the TE in the permuted data (in which case it is a unique contribution).

We note that the difficulties mentioned under 2. and 3. are generic and do not apply to the analysis of transfer entropy alone but to any spectrally-resolved measure of information transfer. They simply reflect the complex nature of statistical dependencies in multi-variable systems.

**Comparing population statistics of frequency-resolved information transfer from a PID perspective.** Somewhat more formally, there are (at least) two options for computing a frequency-resolved measure (population statistic) of information transfer in the form of information theoretic quantities. If we use the notation $X(f)$ for the isolated spectral component at frequency $f$ of process $X$—however this is defined and achieved–, then the approach using an isolation of frequency components before $mTE$-computation aims to compute $I(X^-(f) : Y^+(f') | Y^-(f'))$, i.e. to describe the information transfer from a frequency $f$ in the source to the frequency $f'$ in the target. Here the superscripts $^+$ and $^-$ indicate the history and the present states of the processes. Since all other frequencies have been removed from this expression, redundant information transfer from the set of other frequencies $\{f^\dagger\} \setminus \{f\}$ for the source and the set $\{f^\dagger\} \setminus \{f'\}$ for the target will appear again at other frequencies. In a PID perspective this approach lumps unique information transfer from $f$ to $f'$ together with any redundant information transfer from or to frequencies other than $f, f'$. Our approach (algorithm I) differs first by dropping the frequency resolution for one of the processes. If we take the source perspective, we thus replace $Y^+(f')$ and $Y^-(f')$ with $Y^+$ and $Y^-$. Next, since we are looking for the frequency specific drop which arises from destroying unique synergistic contributions of the

source frequency $f$, but not from destroying redundant contributions, we measure the sum of unique and synergistic contributions. As per the consistency equations of PID [44] this sum is the conditional mutual information $I(X^-(f): Y^+|Y^-, X(\{f^\dagger\} \setminus \{f\}))$. In the SOSO algorithm II, we correspondingly aim to measure $I(X^-(f): Y^+(f')|Y^-, Y^+(\{f^\dagger\} \setminus \{f'\}), X^-(\{f^\dagger\} \setminus \{f\}))$, i.e. frequency resolution for the target is added, as well as an additional conditioning removing redundancies in the target. In sum, the 'classic' approach of isolating spectral components first focuses on all information transferred by a spectral component. In contrast, our approach focuses on the information transferred specifically by that spectral component, possibly in conjunction with others (but not redundantly with others). Apart from the practical problems associated with the classic approach, the choice of one of the two approaches is foremost a matter of the scientific question at hand. The PID framework allows to formulate this question more precisely than possible previously.

**Relation of the partial-information decomposition framework and the SOSO-algorithm.** Understanding that measuring frequency-resolved TE is a PID-problem is particularly useful in understanding the necessity of the SOSO-algorithm to determine putative cross-frequency effects (see *Algorithm II: Testing for direct information transfer from source to receiver frequencies*). For system A in Fig 2 there is direct information transfer from one source to one target frequency. Algorithm I will identify these frequencies and the SOSO-algorithm will confirm the direct transfer of information between these frequencies. In system B, however, one source frequency sends information to all target frequencies except one. This one target frequency, in turn, receives other information from all source frequencies except the identified one. In this system the information sent redundantly by multiple frequencies, and the other information received redundantly will not be revealed, as destroying individual frequencies does not destroy it (information is present redundantly in other frequencies). What algorithm will indicate is the information transfer non-redundantly emitted from one source frequency and (another) information transfer non-redundantly received by one target frequencies. Yet, the SOSO-analysis will show correctly that the identified source and target frequency do not exchange information directly. In system C the same source frequency sends information redundantly into all target frequencies, while one target frequency receives information redundantly from all source frequencies. Depending on the signal to noise ratio, here either no source and no target frequency will be identified by algorithm I, as all the information is carried redundantly, or both the source and the target frequency will be detected if they have superior signal to noise ratio. Then, the direct transfer from one source to one target frequency will be confirmed by the SOSO-algorithm.

Again, the difficulties that arise in the identification of sender and receiver frequencies and in establishing a direct transfer of information from one source frequency to one target frequency are closely related to the fact that we deal with a partial information decomposition problem.

## Methodological advantages and drawbacks of the proposed method

The most important advantage of obtaining frequency-selectivity without filtering the original data is that we do not introduce distortions into the relative timing of the original data or have to deal with other filtering-related artifacts, or even ineffective filters as they were described before. Ultimately this protects us from generating false positive results due to filtering.

With respect to the influence of filtering artifacts biasing TE estimation, as described in [5], manipulating only the surrogate data in the spectral domain restricts the appearance of filtering-related artifacts to the surrogate data. If these artifacts will produce an erroneous increase in TE then this will only lead to conservative errors; if artifacts produce a decrease, this is the

desired effect in surrogate data creation, anyways. We consider these filtering-related errors to be mild in most cases, as demonstrated by the correct recovery of relevant source and target frequencies of information flows in our evaluation examples.

Our approach also solves the problem of filter ineffectiveness as described in [7]. This problem arises in approaches that use filtering of the original data to remove spectral power in frequencies of no interest. Yet, filtering will only dampen spectral power, not remove the information contained in a specific frequency as long as the numeric resolution is high enough to keep the unwanted signal above the numeric quantization noise. Ironically, this problem is more serious when using high-precision math libraries. In our approach we use the filtering-equivalent MODWT transform only to *isolate* the information of interest in order to then destroy the information by coefficient scrambling in surrogate data creation.

In terms of drawbacks, the most important one is likely to be the computational burden our analyses presents, especially when using the SOSO-algorithm. This burden stems from the use of information theoretic estimators for continuous data as well as from recursively nested non-parametric statistics with sufficient iterations and permutations. Due to the computational burden, our method does not lend itself to large-scale exploratory studies as they have been popular with simpler methods based on correlations for example. We therefore advise to apply the methods presented here in a confirmatory way, e.g. for testing highly specific hypotheses of interest, or to neural interactions that have been carefully pre-selected by other methods (e.g. by an mTE analysis, or simply by drawing on prior knowledge).

On the other hand, the hierarchical approach of first searching for source and target frequencies of interest—and only then applying the SOSO-algorithm to selected frequency pairs in a confirmatory step—makes our methods scale better with the number of frequencies involved (basically $\mathcal{O}(j)$ instead of $\mathcal{O}(j^2)$, where $j$ is the number of frequencies involved).

Last, we note, that the frequency resolution that our methods provides is at the level of wavelet scales. This means that we ascribe the information coming from the source and the information reaching the target to specific frequency bands (i.e. the wavelet scales), not single frequencies.

## Specific caveats

The advantage of the proposed algorithms of avoiding filtering of the original data comes at the price that the measurement of the frequency-specific TE contribution is not in absolute terms, but as a difference to the TE value obtained from the surrogate data. This will lead to a potential underestimation of the information transferred by a certain frequency. Yet, for finite data, a comparison of estimated TE values to the TE obtained from suitable, case-specific surrogate data is recommended in most cases anyway, due to the considerable bias problems inherent in the estimation of information theoretic quantities from finite data [33]. Moreover giving exact quantities for the information sent or received by a frequency again means having to face questions of how to attribute uniquely, redundantly or synergistically sent or received information.

## Nonstationary signals

To reliably estimate TE, an assumption of stationarity is required. In cases where such an assumption cannot be made, exploiting shorter time windows or observing multiple realizations (i.e. an ensemble) of the system might alleviate this problem [47]. If the *mTE* algorithm wrongly identifies the interaction in the system under analysis due to the nonstationary properties of the signals, this is reverberated also in the spectral *mTE* analysis. One possible way to alleviate these problem due to nonstationarities is to keep the relevant nonstationarities intact

also in the surrogate data. If one has to deal with a nonstationary processes (e.g. stochastic processes with time-varying mean or variance), one way to achieve this is to sample replications (epochs) from these processes such that these replications all start at the same random variable of the process. Surrogate data for network reconstruction and computation of $mTE_{tot}$ can then be obtained by exchanging full replicas/epochs (this is done in $IDT^{xl}$, for example); correspondingly, for estimates of the spectrally-resolved information transfer the full unaltered sequences of wavelet coefficients at the desired frequency are exchanged between epochs. This way, any difference between original measures and those obtained from surrogate data do not arise from the nonstationarities that have been kept intact in the surrogate data.

## Why are the values for spectral mTE sometimes systematically elevated for the surrogate data?

At first it may seem an indication of a problem of our method that the distribution of values for mTE from surrogate data is sometimes systematically higher than for the actual data. This is however, just an expression of the problems of filtering mentioned in the introduction, i.e. it reflects that filtering can create artefactual increases in TE. However, while in standard approaches this leads to false positives, in our approach it makes the method more conservative. We also note that some choices of the wavelet and permutation methods can be made to alleviate this problem (see S3 Fig). At the moment it is an open question whether there is one set of parameters for surrogate data creation that is optimal for all cases.

## Relation to previous approaches

While we have seen attempts at frequency resolved *TE* and *mTE* estimation at conferences, none of these seem to have been published. The literature on frequency-resolved *TE* is thus very sparse. Specifically, we found that the approaches in [3, 48–50] all use spectral decomposition techniques on the original data, and thus seem suffer from the vulnerabilities detailed in [5, 7]. As one exception, Xu and colleagues [51] use a technique similar to the definition of spectral Granger causality, but for this approach to work they rely on Gaussianity of the data. Some important progress on filtering-related problems has however been made recently for linear multivariate auto-regressive processes using state space formalisms (see [52] and references therein) and Granger Causality. Yet, at present these approaches seem to be focused on within-band (or -scale) information transfer. Thus, they have a narrower focus compared to our problem setting. However, if a system of interest is well described by a multivariate linear auto-regressive process, and exhibits (only) within-band information transfer the methods from [52] will be more data-efficient and come with a much lower computational burden.

With respect to measures releated to transfer entropy proper, we note that the phase transfer entropy introduced by Lobier [50] as the transfer entropy between the time series of the instantaneous signal phases extracted by the narrow band filtering and application of the Hilbert transform is conceptually very different from a our approach, and from TE in general, as it reduces the dimensions of state spaces of the variables entering the TE computation to just two, thereby potentially hiding interesting aspects of the system under study.

Last, the recent approach of Wan [53] is similar to ours in that it uses a wavelet decomposition of the data and a permutation approach to create surrogate data (for bias correction). It differs from our approach however, in that it does not decouple multivariate network identification from spectral decomposition, and most importantly the spectral techniques are applied to the original time series, rather than to the surrogate data only. Thus, their technique can still be seen as a filtering-based approach. Also, the investigation of cross-frequency information transfer is explicitly left for future refinements of their method.

Another important difference of the approach proposed here and previous approaches is the recognition of the problem of frequency resolved TE as a PID-problem—warranting separate analyses for sender and receiver and a particular care in the interpretation of results as laid out above.

When dependencies between different frequencies are of interest, it is sometimes the envelope of a faster frequencies that carries the information to be transferred and that influences other spectral components. To detect such phenomena, often spectral power envelopes are explicitly extracted via narrow-band filtering and a subsequent Hilbert-transform (e.g. [54, 55]), or simply via rectification of a band limited signal and low pass filtering of the rectified signal [3]. These envelopes are then subjected to some technique to compute information transfer, such as TE or GC. We note that such preprocessing steps should not be necessary when using transfer entropies and surrogate data creation, because the time varying aspects of the envelope will be absent in the surrogate data with scrambled phases of the carrier signal, and thus the information encoded in the envelope will be gone. This in turn leads to desired drop in information transfer at the carrier frequency in the surrogate data when using algorithm I.

## Conclusion

We here present an algorithm that returns the frequencies at which a source sends information to a target via any (possibly nonlinear) mechanism, or at which the target receives information from a source. We discuss that a full analysis of frequency-resolved information transfer is a problem of the partial-information decomposition type, such that results should be interpreted carefully, and in the light of possible synergies and redundancies between frequencies in the source or the target. Against this background, we also present a test for a potential one-to-one information transfer from a source frequency to a target frequency. While our algorithms are motivated by problems from neuroscience they are applicable in all fields where frequency-specific information transfer is of interest, e.g. turbulence or climate research.

Our method is fully available and integrated in the open source package IDTXl: https://github.com/pwollstadt/IDTxl/tree/feature_spectral_te, along with a demo script.

## Supporting information

**S1 Fig. Spectrally resolved transfer entropy for example 8 with LA16.** See Fig 4 for display conventions. (Left column) Information transfer, drops when wavelet coefficients are selectively shuffled at scale 5 (frequency band 4-8 Hz) on the source site. The corresponding reception of information at the target is shown on the right column, where a drop for shuffled wavelet coefficients is observed at scale 1 (frequency band 63-125 Hz), scale 2 (frequency band 31-63 Hz) and scale 3 (frequency band 16-31 Hz).
(TIFF)

**S2 Fig. Spectrally resolved information transfer between MEG sources when preparing to detect houses.** See Fig 4 for display conventions. Spectrally resolved information transfer between aIT as a source and FFA as a target in the condition where subjects are trying to detect target houses. aIT sends information mainly at 75-150Hz (left column), whereas FFA receives information at high frequencies (75-150Hz and above) as well as low frequencies (9-19Hz and 5-9Hz) (right column).
(TIFF)

**S3 Fig. Spectrally resolved transfer entropy for example 6 with IAAFT surrogates.** See Fig 4 for display conventions. (Left column) At the source site the maximum drop of the wavelet

coefficients is at scale 5. (Right column) The distributions of surrogates at the target site exhibit less increase compared to the ones in Fig 10B, obtained with the block resampling method, especially at lower scales.
(TIFF)

## Acknowledgments

We are grateful to Christopher J. Keylock for introducing the original idea of combining TE with IMODWT-generated surrogate data to us, and for fruitful discussions on the topic.

## Author Contributions

**Conceptualization:** Michael Wibral.

**Data curation:** Edoardo Pinzuti, Patricia Wollstadt, Michael Wibral.

**Formal analysis:** Edoardo Pinzuti.

**Funding acquisition:** Oliver Tüscher, Michael Wibral.

**Investigation:** Edoardo Pinzuti.

**Methodology:** Edoardo Pinzuti, Patricia Wollstadt, Aaron Gutknecht, Michael Wibral.

**Project administration:** Oliver Tüscher, Michael Wibral.

**Resources:** Oliver Tüscher, Michael Wibral.

**Software:** Edoardo Pinzuti, Patricia Wollstadt, Michael Wibral.

**Supervision:** Oliver Tüscher, Michael Wibral.

**Validation:** Edoardo Pinzuti, Patricia Wollstadt, Aaron Gutknecht.

**Visualization:** Edoardo Pinzuti.

**Writing – original draft:** Edoardo Pinzuti, Michael Wibral.

**Writing – review & editing:** Edoardo Pinzuti, Patricia Wollstadt, Aaron Gutknecht, Oliver Tüscher, Michael Wibral.

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
