## [Decision Letter · Decision Letter 0]

22 Jun 2020

Dear Mr Pinzuti,

Thank you very much for submitting your manuscript "Measuring spectrally-resolved information transfer" for consideration at PLOS Computational Biology.

As with all papers reviewed by the journal, your manuscript was reviewed by members of the editorial board and by several independent reviewers. In light of the reviews (below this email), we would like to invite the resubmission of a significantly-revised version that takes into account the reviewers' comments.

We cannot make any decision about publication until we have seen the revised manuscript and your response to the reviewers' comments. Your revised manuscript is also likely to be sent to reviewers for further evaluation.

Sincerely,

Daniele Marinazzo

Deputy Editor

PLOS Computational Biology

Daniele Marinazzo

Deputy Editor

PLOS Computational Biology

Reviewer's Responses to Questions

**Comments to the Authors: **

Reviewer #1: Summary

-------

This study proposes an approach to analysis of cross-frequency information transfer in complex processes constituted by multiple dynamically-interacting elements, particularly neural systems. It is well-argued and exceptionally clearly presented. The relevant literature appears to have been thoroughly researched.

The authors identify the issue that more conventional frequency analyses of information transfer (such as spectral Granger causality) are confined to within-band information transfer, whereas information transfer is potentially many-to-many across the frequency spectrum. The authors correctly recognise that this problem cannot be addressed simply by pre-filtering, and present a solution which uses the maximum overlap discrete wavelet transform (MODWT) to construct surrogate data to instantiate appropriate null distributions for statistical inference. This procedure is validated on synthetic "ground-truth" data, and applied to some neurophysiological datasets, with novel results.

General points

--------------

* While acknowledging that this is common practice in some disciplines, to my mind the language of complex, especially neural, processes, as performing "computation" sits a little uneasily, especially with a neuroscience audience. "Computation" (to me at any rate) suggests an end-to-end algorithm with known input, output and intent. From a neuroscience perspective in particular, the more neutral term "function" (cognitive, behavioural, etc.) feels more appropriate. Not a big deal, though.

* One concern I have is the emphasis on putative partial information decomposition (PID); I say "putative", since, as the authors fully acknowledge, no entirely satisfactory PID has been found to date. More worryingly, it is not even clear that satisfactory PIDs actually exist at all, less so PIDs which, like Shannon (mutual/conditional) information are invariant under transformation of (continuous-state) variables - i.e., how quantities are measured. In my opinion, bringing PID into the picture presents something of a distraction, and potentially a hostage to fortune; this work stands its ground without reference to (possibly chimerical) PIDs.

* One point that should be clarified, is the extent to which "side effects" of the surrogate construction might introduce statistical artefacts. Suppose,e.g., we have a three-variable "indirect" information transfer structure X -> Y -> Z (but not directly X -> Z); i.e., so that T(X -> Z) > 0, but T(X -> Z | Y) = 0. If we wish to discover by the surrogate method whether source frequency-specific T(X -> Z | Y) is significantly different from zero, we would presumably construct (frequency-specific) surrogates for the source X. But in doing so, we also inadvertently impose source frequency T(X -> Y) = 0. Might this confound inference on T(X -> Z | Y)? Similarly for target frequency TE and the "SOSO" procedure. It seems to me that the synthetic examples supplied do not address this situation, or more generally the interaction of surrogate construction with conditioning, which is a necessity for principled analysis of direct information transfer in multivariate scenarios.

* Not to be taken as a criticism, but an interesting omission in this study is an analytic, quantitative description of cross-frequency TE; the study presents purely an inferential procedure. Conventional TE (and, e.g., spectral Granger-Geweke causality) have appealingly intuitive quantitative population statistics, which not only may be tested in sample, but stand as "effect size" for information transfer (clearly the authors do appreciate this, when they say "The advantage of the proposed algorithms of avoiding filtering of the original data comes at the price that the measurement of the frequency-specific TE contribution is not in absolute terms, but as a difference to the TE value obtained from the surrogate data"). But does that price need to be paid? Do the authors believe population statistics exist for their cross-frequency TE?

* I would like to see some discussion of the impact (if any) of non-stationarity on the method.

* An alternative (although not directly comparable) potential "cross-frequency" approach is via the amplitude (not phase) envelope constructed from the analytic signal of a band-limited signal. This has been, e.g., applied to ECoG data, where there is good evidence that the high-frequency envelope correlates well with average neural firing rates. See e.g., Norman et al., Nature Communications 8(1301), 2017. Perhaps some discussion could be included on this.

Specific points

---------------

* p 4, para 2: The authors note that calculation of mTEtot is NP-hard. It may be worth remarking that in practice the situation is even more vexed, since (especially in neural systems) we rarely have access to all potentially-relevant exogenous/latent information sources, that these sources are likely to be far too numerous to account for even in principle, and that measured sources/targets are likely to be spatio-temporal aggregates of "real" (physical) sources.

* p 17, penultimate para: Since (sample) TE cannot be assumed to be normally distributed, is a t-test appropriate for testing mTEtot, or might it be more appropriate to use a nonparametric test (e.g., Mann-Whitney, or some such)?

Reviewer #2: This work proposes a new method to identify frequency-specific information transfer between coupled signals, based on comparing transfer entropy (TE) measures computed on two original signals and on surrogate signals obtained destroying the oscillatory structure within specific frequency bands. The method is tested in several simulation settings, and applied to MEG and LFP recordings.

Main comments:

- The Authors present as a main advantage of the proposed method the fact that the original signals are unfiltered before computing the information transfer. Indeed, it is known that filtering can have adverse effects on the computation of information transfer and more generally Granger-causal (GC) measures of directed coupling (refs 5-7). However, filtering is applied at the level of surrogate data to rule out some specific temporal scales from the signals prior to computation of the same TE measure applied to the original signals, and inferences about frequency specific directed coupling are made based on the comparison between original and surrogate signals. Therefore, since filtering is actually exploited to make inferences about information transfer, it is not very well clear how the related problems are circumvented in the analysis proposed here.

- In relation to the comment above, it is known that linear Granger causality is theoretically invariant under filtering, and therefore differences between original and filtered GC measures are due to estimation problem (e.g., ref. 7). Thus, the question arises about what are the mechanisms that allow to reveal “real” (i.e. not due to filtering effects in the application to surrogates) information transfer in this TE application: can they be related to nonlinearities captured by the model free approach, or by the use of the MODWT method for surrogate generation?

- In my understanding, the proposed method allows to detect whether significant information transfer is present between specific time scales in two signals, but does not test the significance of the information transfer against the zero level. To clarify this issue, it would be appropriate to state explicitly the null hypothesis against which the different algorithms are tested, and to report whether and how the evaluation of statistically significant information transfer has been performed in both simulations and real data applications.

- Simulations exhaustively test the frequency-specific information transfer in situations of unidirectional coupling between two signals. However, simulations are not reported about bidirectionally coupled signals, nor regarding multivariate signals where typical confounders (e.g., common drivers, cascades) are introduced. It would be desirable to see how the proposed method behaves in these settings.

- It would be also good to report more details about the estimation of TE in terms of description of the estimator chosen, setting of its specific parameters, and about the data requirement (appropriate length of the time series which allows reliable TE estimation)

- In the simulation of coupled Roessler systems, no significant drop was observed for the TE at the target site; however, in Fig. 10 it seems that the TE is significantly below the surrogate distribution. How would you interpret this apparently higher information transfer on the surrogates?

- In the final part of the paper where the relation to previous approaches is discussed, the Authors may want to discuss also the conceptual differences of their method with the linear and model free measures of multiscale Granger causality and information transfer introduced in [Faes, L., et al. (2017), Physical Review E, 96(4), 042150] and in [Wan, X., & Xu, L. (2018), PloS one, 13(12), e0208423].

Technical/minor:

- The simulation figures are probably too many; a condensed representation of the simulation results (e.g., one figure per simulation) could be envisaged; the same may hold for the second application on real data.

- The time t should appear in the cosine terms of eqs. 10-11, 17-18, and others.

Reviewer #3: This work introduces a method for identifying spectral band specific transfer entropy based on the statistical tests with the surrogate data from discrete wavelet transforms. The method involves five steps of computation, relies on several free parameters (e.g. time lag and number of discrete wavelet components), and is computationally expensive. However, the method is shown to uncover interesting insights into the dynamics of frequency-specific interactions. The validity of the method is demonstrated with some synthetic data generated from coupled systems of sinusoidal oscillators and time-delay coupled nonlinear oscillators. The method is also applied to real data (human MEG and ferret lfps). The main novelty of the method is the combination of the transfer entropy measure and frequency-specific surrogate data testing, which is claimed to avoid filtering of the original time series. The method has not be compared with any other existing techniques even for linear systems. I have the following comments and questions.

1. The method needs to be tested first in an uncoupled system of oscillators to show how the transfer entropy behaves with wavelet data surrogates. 

2. How does the method work with the data from a system of interacting AR models with single frequency peak at the source and/or target? Are the results comparable to the results from spectral Granger causality method? 

3. The wavelet method is used to uncover time-varying spectral features. The proposed work seems to focus on stationary (non-time varying) processes. 

4. This work advocates against filtering of original signals. However, there are no or little issues with filtering producing artifacts if appropriately applied to long time series. These results (at least from a simple model of sinusoidal oscillators) need to be compared with the transfer entropy results obtained from filtered data in some of the simple models. 

 I cannot recommend the manuscript in the present form for publication.

**Have all data underlying the figures and results presented in the manuscript been provided?**

Reviewer #1: Yes

Reviewer #2: Yes

Reviewer #3: Yes

PLOS authors have the option to publish the peer review history of their article (what does this mean?). If published, this will include your full peer review and any attached files.

Reviewer #1: Yes: Lionel Barnett, Sackler Centre for Consciousness Science, Department of Informatics, Sussex University, UK

Reviewer #2: Yes: Luca Faes

Reviewer #3: No
---

## [Decision Letter · Decision Letter 1]

9 Oct 2020

Dear Mr Pinzuti,

Thank you very much for submitting your manuscript "Measuring spectrally-resolved information transfer" for consideration at PLOS Computational Biology. The reviewers and the editors appreciated the resubmission. Based on the reviews, we are likely to accept this manuscript for publication, providing that you modify the manuscript according to the review recommendations still outstanding. 

Sincerely,

Daniele Marinazzo

Deputy Editor

PLOS Computational Biology

Daniele Marinazzo

Deputy Editor

PLOS Computational Biology

[LINK]

Reviewer's Responses to Questions

**Comments to the Authors: **

Reviewer #1: I would like to thank the authors for their considered responses to the points raised in my initial review. I am happy to recommend publication,

Reviewer #2: The Authors have addressed thoroughly and satisfactorily all concerns raised. The revised manuscript is a substantial improved version of the original work, that was already novel and of good quality.

Reviewer #3: -The authors make claim that spectral Granger causality is not able to uncover the causal influences in the case of cross-frequency coupling. In their examples, they used the parametric approach to Granger causality for the case of unidirectional coupling (Figure 6), did not use that approach for the case of cross-frequency coupling but instead used the nonparametric approach to show the negative result (Figure 7). There are additional steps (e.g. multi-taper method) needed for the nonparametric approach to create the spectral matrix. Those details are missing. I do not understand why the same parametric (or nonparametric) approach was not used for both examples to compare.

- The method cannot necessarily pinpoint a specific frequency in directed connectivity but can identify a frequency range, the extent of which is not discussed in the manuscript. This limitation needs to be discussed in the main text.

**Have all data underlying the figures and results presented in the manuscript been provided?**

Reviewer #1: Yes

Reviewer #2: Yes

Reviewer #3: Yes

PLOS authors have the option to publish the peer review history of their article (what does this mean?). If published, this will include your full peer review and any attached files.

Reviewer #1: No

Reviewer #2: **Yes: **Luca Faes, Dept. of Engineering, University of Palermo, Italy

Reviewer #3: No
---

## [Editor Report · Decision Letter 2]

12 Nov 2020

Dear Mr Pinzuti,

We are pleased to inform you that your manuscript 'Measuring spectrally-resolved information transfer' has been provisionally accepted for publication in PLOS Computational Biology.

Best regards,

Daniele Marinazzo

Deputy Editor

PLOS Computational Biology

Daniele Marinazzo

Deputy Editor

PLOS Computational Biology

---

## [Editor Report · Acceptance letter]

18 Dec 2020

PCOMPBIOL-D-20-00853R2 

Measuring spectrally-resolved information transfer

Dear Dr Pinzuti,

I am pleased to inform you that your manuscript has been formally accepted for publication in PLOS Computational Biology. Your manuscript is now with our production department and you will be notified of the publication date in due course.

With kind regards,

Livia Horvath
